# CAN WE PREDICT PERFORMANCE OF LARGE MODELS ACROSS VISION-LANGUAGE TASKS?

## ABSTRACT

Evaluating large vision-language models (LVLMs) is very expensive, due to the high computational costs and the wide variety of tasks. The good news is that if we already have some observed scores, we may be able to infer unknown ones. In this study, we propose a new framework for predicting unknown performance scores based on observed ones from other LVLMs or tasks. We first formulate the performance prediction as a matrix completion task. Specifically, we construct a sparse performance matrix $R$, where each entry $R_{mn}$ represents the performance score of the $m$-th model on the $n$-th dataset. By applying probabilistic matrix factorization (PMF) with Markov chain Monte Carlo (MCMC), we can complete the performance matrix, that is, predict unknown scores. Additionally, we estimate the uncertainty of performance prediction based on MCMC. Practitioners can evaluate their models on untested tasks with higher uncertainty first, quickly reducing errors in performance prediction. We further introduce several improvements to enhance PMF for scenarios with sparse observed performance scores. In experiments, we systematically evaluate 108 LVLMs on 176 datasets from 36 benchmarks, constructing training and testing sets for validating our framework. Our experiments demonstrate the accuracy of PMF in predicting unknown scores, the reliability of uncertainty estimates in ordering evaluations, and the effectiveness of our enhancements for handling sparse data.

## 1 INTRODUCTION

It is expensive to evaluate large vision-language models (LVLMs). First, large-scale models result in significant computational or API calling costs and memory usage. Additionally, since a single LVLM can handle a wide range of tasks, comprehensively understanding model performance on different tasks becomes more challenging. As a result, hundreds of benchmarks have been proposed to assess the strengths and weaknesses of LVLMs (Li & Lu, 2024). Zhang et al. (2024b) report that it takes hundreds of hours to evaluate one model on around 50 tasks in LMMs-Eval, and evaluation even exceeds 1,400 hours on models of 100B parameters or more.

Fortunately, we have already observed performance scores from some of these models on some tasks, for instance, from the official reports of released models and datasets. For new models, scores can also be readily obtained with limited compute by running on a small number of tasks. If these observed scores can be used to predict unknown ones, we could avoid unnecessary evaluations and effectively reduce costs. Recent works (Polo et al., 2024; Zhang et al., 2024b) require running the same model on the same task to predict model performance, and most of them ignore the potential of leveraging observed performance data from other models or tasks.

In this study, we propose a new framework for predicting unknown performance scores based on observed ones from other LVLMs or tasks. We first formulate this as a matrix completion problem. Specifically, we construct a sparse performance matrix $R$ where each entry $R_{mn}$ represents the performance score of the $m$-th model on the $n$-th dataset. By applying probabilistic matrix factorization (PMF) with Markov chain Monte Carlo (MCMC), we can predict unknown performance scores based on observed entries in the matrix. A summary of the framework is shown in Fig. 1.

A bonus of our framework is active evaluation, which aims to select a subset of model-dataset pairs to evaluate in order to minimize prediction errors across the entire performance matrix. Given a PMF model on a very sparse performance matrix, we calculate prediction uncertainty from MCMC

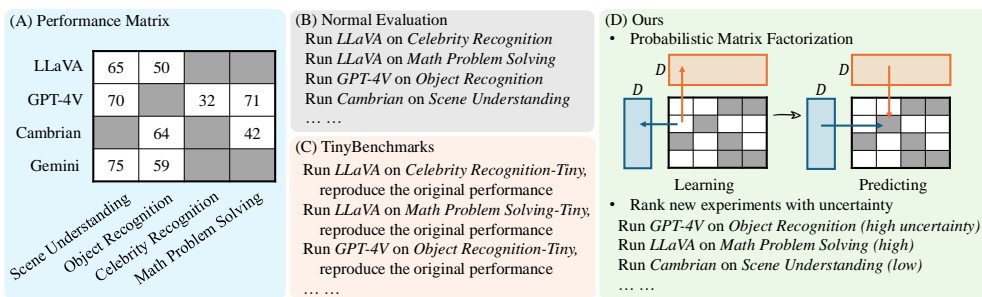

Figure 1: **Framework**. (A) Given a sparse matrix of performance scores of LVLMs on various tasks, the goal is to estimate the missing entries. (B) A normal way is to evaluate untested model-dataset pairs one-by-one. (C) TinyBenchmarks (Polo et al., 2024) runs models on smaller test sets and reproduce the original performance. (D) We use Probabilistic Matrix Factorization (PMF) to predict missing entries, reducing unnecessary evaluations, and rank new experiments based on uncertainty.

and prioritize evaluating model-dataset pairs with high uncertainty. Our experiments will confirm the effectiveness of this strategy for active evaluation.

A challenge is that PMF tends to predict the average score for models and datasets with very few observed scores, resulting in poor prediction results (Mnih & Salakhutdinov, 2007). To address this, we introduce several improvements to enhance PMF for scenarios with sparse observed data. First, we extend PMF to a simple tensor factorization approach, which can handle multiple performance metrics across different vision-language tasks. Second, we utilize Bayesian PMF (Salakhutdinov & Mnih, 2008) with an LKJ prior (Lewandowski et al., 2009) on the variance. Third, we also incorporate extra information as model and dataset profiles to improve performance prediction. For example, if we know a model uses CLIP as a vision encoder, the information may help predict the model's performance, especially when we observe only a few performance scores of the model.

In experiments, we conduct a systematic evaluation of 108 LVLMs across 176 distinct datasets derived from 36 existing benchmarks, based on four prior works (Duan et al., 2024; Zhang et al., 2024b; Liang et al., 2024; Karamcheti et al., 2024). We evaluate open-source models such as LLaVA-v1.5 (Liu et al., 2023a), InstructBLIP (Dai et al., 2023), mPLUG-Owl (Ye et al., 2023), and MiniGPT-4 (Zhu et al., 2023), as well as closed-source models including GPT-4o, GPT-4 (Achiam et al., 2023), Gemini-1.5 (Reid et al., 2024). The benchmarks cover general VQA (Li et al., 2023a), knowledge-dense VQA (Yue et al., 2024), hallucination (Li et al., 2023b), medicine (He et al., 2020), emotion recognition (Goodfellow et al., 2013), and others. To reduce computational and API costs, we subsample some datasets, following the practice in Liang et al. (2024).

Using the results from 108 LVLMs across 176 datasets, we construct a $108 \times 176$ performance matrix, with some entries masked for testing. We empirically demonstrate that PMF accurately predicts masked scores and consistently outperforms baselines as long as more than 10% entries in the performance matrix are observed. We also show that selecting high-uncertainty model-dataset pairs for evaluation significantly reduces prediction errors compared to random selection. Additionally, our improvements effectively alleviate the sparse data issue of PMF.

In summary, this paper covers three main points. First, we formulate a problem of predicting the unknown performance of LVLMs across tasks. Second, we apply the well-established PMF algorithm to this problem, show the application of active evaluation, and propose several strategies to mitigate the sparse data issue. Third, we conduct a comprehensive evaluation of 108 LVLMs across 176 datasets, constructing training and testing sets for further experiments.

## 2 RELATED WORKS

### 2.1 RECENT LVLMS AND BENCHMARKS

In recent years, there has been increasing growth in LVLMs, with many new models demonstrating impressive capabilities. Notable closed-source models include GPT-4 (Achiam et al., 2023) and

Gemini (Team et al., 2023), while open-source models such as LLaVA (Liu et al., 2024; 2023a), InstructBLIP (Dai et al., 2023), and InternVL (Chen et al., 2023; 2024) have also gained widespread attention. Karamcheti et al. (2024) explore the design of LVLMs and have released a series of models (i.e., Prismatic VLMs) featuring different architectures and training strategies.

These LVLMs can handle a wide variety of tasks within a single model, but this versatility also requires more various benchmarks to fully understand their strengths and weaknesses. Some existing benchmarks can be repurposed for assessing these models, such as Flickr30k (Young et al., 2014), GQA (Hudson & Manning, 2019), and OKVQA (Marino et al., 2019). Recent works also propose new benchmarks to evaluate LVLMs in handling dense knowledge, complex reasoning, and decision-making tasks. Examples of novel benchmarks include SEED-Bench-2 (Li et al., 2023a), MMMU (Yue et al., 2024), and MME (Fu et al., 2023). Additionally, as LVLMs become more integrated into everyday applications, benchmarks like POPE (Li et al., 2023b) have been introduced to assess trustworthy issues like hallucination in these models. The variety of LVLMs and benchmarks leads to substantial computational demands and memory usage.

## 2.2 IMPROVE EVALUATION EFFICIENCY

Recent works introduce unified frameworks to assess models across multiple benchmarks using a single codebase, such as VLMEvalKit (Duan et al., 2024), LMMs-Eval (Zhang et al., 2024b), and HEMM (Liang et al., 2024). Our study builds on these efforts by consolidating their evaluation frameworks and integrating models in Prismatic VLMs series.

Predicting unknown model performance can reduce the evaluation cost. Recent works select a core-set of samples from a large benchmark, for evaluating LLMs (Polo et al., 2024; Perlitz et al., 2023) and LVLMs (Zhang et al., 2024b; Zhu et al., 2024). The performance of a specific model on the core-set is used to estimate its performance on the full benchmark. Besides, prior studies estimate model performance on an unlabeled test set based on distribution shift (Deng & Zheng, 2021), confidence scores (Guillory et al., 2021; Yang et al., 2024), or LLM feedback (Zheng et al., 2023). Instead of running models on a coreset or an unlabeled set, our framework predicts unknown performance by utilizing the correlation between model performances across benchmarks.

Another related direction is adaptive testing (Rodriguez et al., 2021; Prabhu et al., 2024). Given a new model, only a subset of samples is selected based on sample difficulty for evaluating the new model. While their work focuses on sample-level testing with a single metric, our approach operates at the dataset level, using six different metrics. Furthermore, instead of relying on statistically inferred sample difficulty, we propose a method to rank model-dataset pairs for evaluation based on uncertainty in performance prediction from MCMC.

## 2.3 PROBABILISTIC MATRIX FACTORIZATION

PMF (Mnih & Salakhutdinov, 2007) is a technique widely applied in recommender systems. Given part of the ratings that users provide for items, the goal is to model the observed ratings and predict the missing ones. PMF achieves this by decomposing the observed rating matrix into two lower-dimensional matrices, representing the latent features of users and items. A rating is modeled as a Gaussian distribution centered around the dot product of the user's and item's feature vectors.

One major challenge with PMF is that, if users rate very few items, their predicted ratings will be near the average for those items. Bayesian PMF (BPMF) (Salakhutdinov & Mnih, 2008) addresses this by placing distributions over the priors of the latent user and item features, making it more effective in handling sparse data. Additionally, Constrained PMF (Mnih & Salakhutdinov, 2007) introduces a latent similarity constraint matrix to further refine the user feature vectors.

## 3 MODELLING LVLM PERFORMANCE

In this section, we first describe the application of PMF to model the performance score matrix of LVLMs across datasets. Then, we discuss active evaluation for LVLMs. Last, three techniques are introduced to enhance PMF: supporting multiple metrics, incorporating Bayesian PMF, and integrating model and dataset profiles in modeling.

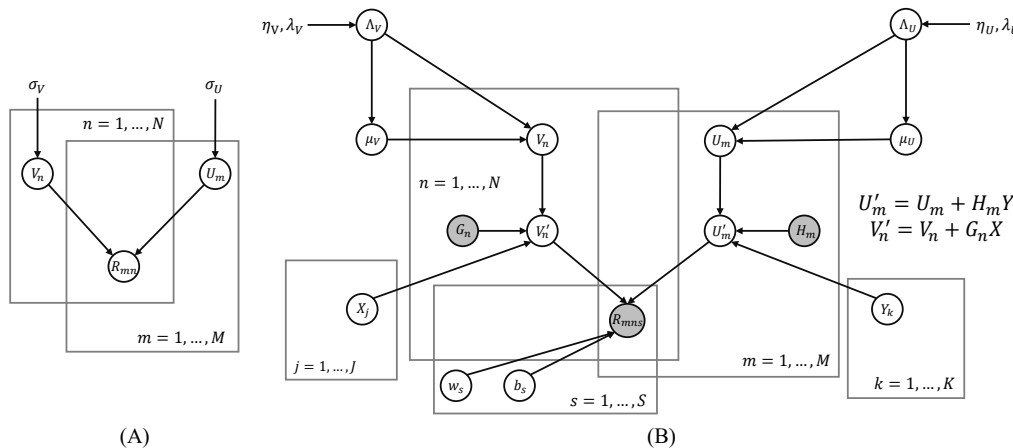

Figure 2: **Graphical Models** of PMF (A) and the enhanced model (B). (A) is adapted from the original paper (Mnih & Salakhutdinov, 2007). In (B), we set the mean to $\mathbf{0}$ and the covariance to the identity matrix, thus omitting most of the hyper-parameters for the random variable distributions.

### 3.1 REVISIT PROBABILISTIC MATRIX FACTORIZATION

Let $\boldsymbol{R}$ be an $M \times N$ matrix representing model performance scores on datasets, where $M$ is the number of models and $N$ is the number of datasets. For simplicity, we initially assume a single performance metric, though in reality, benchmarks often employ multiple metrics. In such cases, $\boldsymbol{R}$ becomes an $M \times N \times S$ tensor, where $S$ represents the total number of metrics. We will address this more complex scenario in the following sections.

In practice, only a subset of the elements in $\boldsymbol{R}$ are observed, meaning we evaluate only a portion of the model-dataset pairs and aim to estimate the remaining performance scores. Specifically, we define a matrix $\boldsymbol{O} \in \{0, 1\}^{M \times N}$, where $O_{mn} = 1$ if $R_{mn}$ is observed, and 0 otherwise.

To model the observed matrix and estimate the unknown values, we employ PMF (Mnih & Salakhutdinov, 2007), as illustrated by the probabilistic graphical model in Fig. 2(A). PMF decomposes $\boldsymbol{R}$ into two low-dimensional matrices, $\boldsymbol{U} \in \mathbb{R}^{M \times D}$ and $\boldsymbol{V} \in \mathbb{R}^{N \times D}$, where $D$ is the latent dimension. Here, $\boldsymbol{U}_{m,:}$ and $\boldsymbol{V}_{n,:}$ are the latent feature vectors for the $m$-th model and the $n$-th dataset, respectively, and we refer to them as $\boldsymbol{U}_m$ and $\boldsymbol{V}_n$. These latent vectors are modeled as multivariate Gaussian distributions, and the observed ratings are assumed to follow a Gaussian distribution centered at the dot product of the latent feature vectors:

$$p(\boldsymbol{R} \mid \boldsymbol{U}, \boldsymbol{V}, \sigma^2) = \prod_{m=1}^{M} \prod_{n=1}^{N} \left[ \mathcal{N} \left( R_{mn} \mid \boldsymbol{U}_m^T \boldsymbol{V}_n, \sigma^2 \right) \right]^{O_{mn}}, \tag{1}$$

$$p(\boldsymbol{U} \mid \sigma_U^2) = \prod_{m=1}^{M} \mathcal{N}(\boldsymbol{U}_m \mid \mathbf{0}_D, \sigma_U^2 \boldsymbol{I}_D), \quad p(\boldsymbol{V} \mid \sigma_V^2) = \prod_{n=1}^{N} \mathcal{N}(\boldsymbol{V}_n \mid \mathbf{0}_D, \sigma_V^2 \boldsymbol{I}_D), \tag{2}$$

where $\boldsymbol{I}_D$ is a $D \times D$ identity matrix, and $\mathcal{N}(x \mid \mu, \sigma^2)$ represents the probability density function of a Gaussian distribution with mean $\mu$ and variance $\sigma^2$. We simply set $\sigma_U = \sigma_V = 1$.

Rather than using Maximum A Posteriori estimation to obtain point estimates of the unknown performance scores in $\boldsymbol{R}$, we apply MCMC to obtain distributions over the estimated scores and quantify the uncertainties in our predictions. Specifically, we use the No-U-Turn Sampler (NUTS) (Hoffman et al., 2014), an advanced Hamiltonian Monte Carlo method (Neal, 2011).

Our experiments show that standard PMF performs well with sufficient observed data. But its performance degrades significantly and is even worse than predicting the mean, when the observed data is very sparse (i.e., fewer than 10% model-dataset pairs are observed). To address this, we enhance our model with several techniques, with a new graphical model shown in Fig. 2(B).

## 3.2 ACTIVE EVALUATION

MCMC allows us to estimate score distributions and readily obtain uncertainty estimates for each unknown score, enabling us to prioritize evaluation experiments. For example, if we are uncertain about GPT-4's performance on a 3D understanding but confident about LLaVA's performance on object recognition, we can prioritize evaluating GPT-4 on the 3D task when our resources are limited.

In our method, we begin by applying PMF to model a sparse performance matrix. Using MCMC, we get hundreds of estimations of each unknown score and calculate the standard deviation of estimations as a measure of uncertainty. The unobserved scores are ranked by their uncertainties. High-uncertainty scores are replaced with ground truth, simulating evaluation process in practice. We rerun PMF with updated observed data, calculate uncertainty, and determine the next set of evaluations. This process is repeated until our resource budget is exhausted or all scores are observed.

## 3.3 MULTIPLE METRICS

Previously, we assumed that each dataset has only one scoring metric, but this is not the case in practice. For example, yes-or-no questions can be evaluated using accuracy, precision, recall, and F1 score, while open-ended questions may use metrics like BART score (Yuan et al., 2021) and BERT score (Zhang et al., 2019). Model performances are represented by a tensor $\mathbf{R} \in \mathbb{R}^{M \times N \times S}$, where $S$ is the total number of metrics. Empirically, we find that using PMF to model and predict each metric independently works well when sufficient data is available. However, when observed data is sparse, incorporating relationships between metrics will be helpful.

To address this, we extend our PMF model into a simple Probabilistic Tensor Factorization (PTF), where we decompose the 3D tensor $\mathbf{R}$ into the product of two low-rank matrices and a 1D vector. This can be interpreted as applying a linear transformation to the original PMF output, translating it into multiple metrics. Specifically, we define:

$$p(\mathbf{R} \mid \boldsymbol{U}, \boldsymbol{V}, \boldsymbol{w}, \boldsymbol{b}, \sigma^2) = \prod_{m=1}^{M} \prod_{n=1}^{N} \prod_{s=1}^{S} \left[ \mathcal{N}\left( \boldsymbol{R}_{mns} \mid (\boldsymbol{U}_m^T \boldsymbol{V}_n) w_s + b_s, \sigma^2 \right) \right]^{O_{mns}}, \qquad (3)$$

$$p(\boldsymbol{w} \mid \sigma_w^2) = \mathcal{N}(\boldsymbol{w} \mid \boldsymbol{0}_S, \sigma_w^2 \boldsymbol{I}_S), \quad p(\boldsymbol{b} \mid \sigma_b^2) = \mathcal{N}(\boldsymbol{b} \mid \boldsymbol{0}_S, \sigma_b^2 \boldsymbol{I}_S), \qquad (4)$$

where we set $\sigma_w = \sigma_b = 1$ for simplicity.

This approach implicitly assumes a linear relationship between scoring metrics, which may not exactly hold in reality. However, we usually observe some linear correlation between the metrics on the same task. Moreover, more sophisticated techniques, such as advanced tensor factorization methods, modeling non-linear metric relationships with neural networks, or using manually defined transformation functions for specific metrics, can be explored to further improve the model.

Note that some metrics may be irrelevant for certain datasets, e.g., accuracy is not meaningful for long-answer questions. While our model can predict these scores, we discard the predicted results.

## 3.4 BAYESIAN PMF

Instead of using fixed priors for the feature vectors, we model the priors using probabilistic distributions, as proposed by Salakhutdinov & Mnih (2008). Unlike the original paper, which employs a Wishart distribution for the variance, we use the LKJ correlation prior (Lewandowski et al., 2009) and an Exponential prior to model the variance, as suggested by the PyMC documentation,

$$\boldsymbol{\Lambda}_U^{-1} = (\mathbf{diag}\,(\boldsymbol{\sigma}_L)\,\boldsymbol{L}_U)(\mathbf{diag}\,(\boldsymbol{\sigma}_L)\,\boldsymbol{L}_U)^T, \qquad (5)$$

where $p(\boldsymbol{L}_U \mid \eta_U) = \mathrm{LKJ}(\boldsymbol{L}_U \boldsymbol{L}_U^T \mid \eta_U)$ and $p(\boldsymbol{\sigma}_L \mid \lambda_U) = \prod_{d=1}^{D} \mathrm{Exp}(\sigma_d \mid \lambda_U)$.

Latent feature vectors are then modeled as:

$$p(\boldsymbol{\mu}_U \mid \boldsymbol{\Lambda}_U^{-1}) = \mathcal{N}(\boldsymbol{\mu}_U \mid \boldsymbol{0}_D, \boldsymbol{\Lambda}_U^{-1}), \qquad (6)$$

$$p(\boldsymbol{U} \mid \boldsymbol{\mu}_U, \boldsymbol{\Lambda}_U^{-1}) = \prod_{m=1}^{M} \mathcal{N}(\boldsymbol{U}_m \mid \boldsymbol{\mu}_U, \boldsymbol{\Lambda}_U^{-1}). \qquad (7)$$

A similar formulation applies to $\boldsymbol{V}$, which we omit here for brevity.

## 3.5 MODEL AND DATASET PROFILES

The final enhancement to our framework is the incorporation of additional information about the models and datasets. For example, knowing that two LVLMs use CLIP as the vision encoder, or that LLaVA-v1.5 and LLaVA-NeXT are developed by the same team, suggests potential relationships in their performances. Inspired by Constrained PMF (Mnih & Salakhutdinov, 2007), we incorporate extra information as model and dataset profiles, to improve performance prediction.

Let $\boldsymbol{H} \in \mathbb{R}^{M \times K}$ and $\boldsymbol{G} \in \mathbb{R}^{N \times J}$ represent the model and dataset profiles, where $\boldsymbol{H}_{m,:}$ encodes $K$ properties of the $m$-th model (e.g., vision encoder type), and $\boldsymbol{G}_{n,:}$ encodes $J$ properties of the $n$-th dataset. We introduce Gaussian-distributed variables $\boldsymbol{Y} \in \mathbb{R}^{K \times D}$ and $\boldsymbol{X} \in \mathbb{R}^{J \times D}$ to learn the effects of these profiles. The latent feature vectors are now the sum of the original vectors and the profile features, following Constrained PMF (Mnih & Salakhutdinov, 2007).

$$p(\boldsymbol{Y} \mid \sigma_Y^2) = \prod_{k=1}^{K} \mathcal{N}(\boldsymbol{Y}_k \mid \mathbf{0}_D, \sigma_Y^2 \boldsymbol{I}_D), \quad p(\boldsymbol{X} \mid \sigma_X^2) = \prod_{j=1}^{J} \mathcal{N}(\boldsymbol{X}_j \mid \mathbf{0}_D, \sigma_X^2 \boldsymbol{I}_D), \quad (8)$$

$$\boldsymbol{U}' = \boldsymbol{U} + \boldsymbol{H}\boldsymbol{Y}, \quad \boldsymbol{V}' = \boldsymbol{V} + \boldsymbol{G}\boldsymbol{X}. \quad (9)$$

**Oracle Profiles.** To explore the upper bound of model and dataset similarities, we use the full $\boldsymbol{R}$ matrix to cluster models and datasets. For each model, we take $\boldsymbol{R}_{i,:}$ (its performance across all datasets) as a vector and apply the K-Means algorithm to cluster all models. We select the optimal number of clusters using the elbow method. Similarly, for each dataset, we cluster $\boldsymbol{R}_{:,j}$ in the same way. We convert the cluster assignments into one-hot vectors to serve as profiles.

**Custom Profiles.** Since oracle profiles rely on complete performance data, they are not practical for real-world use. To overcome this, we define custom profiles that can be applied in practice. For models, we include features such as the number of parameters in the LLM backbone, vision encoder type (one-hot), and the LVLM family (one-hot), illustrated in the supplementary material (Table 4). Additionally, we cluster datasets based on latent representations obtained from various models and get one-hot encoded dataset profiles. We explore three different approaches to generate these latent representations: D1. using MPNet (Song et al., 2020) to encode a short description of each dataset. D2. using CLIP to encode images and BGE-M3 to encode questions in a dataset (following Zhang et al. (2024b)), then averaging the embeddings on the dataset; and D3. using LLaVA-7B to encode both images and text, then averaging the embeddings for the dataset.

## 4 EXPERIMENTS

In this section, we construct a performance matrix and present key experiments for our framework.

### 4.1 EVALUATING MODELS ON BENCHMARKS

Prior works have developed general pipelines for evaluating LVLMs across a wide range of benchmarks (Duan et al., 2024; Zhang et al., 2024b; Liang et al., 2024). Building on these code repositories, we evaluate 108 LVLMs on 36 benchmarks. The open-source models we cover include LLaVA-v1.5 (Liu et al., 2023a), LLaVA-NeXT (Liu et al., 2023a), InstructBLIP (Dai et al., 2023), mPLUG-Owl (Ye et al., 2023), and Prismatic VLMs (Karamcheti et al., 2024). We also evaluate closed-source models such as GPT-4 (Achiam et al., 2023) and Gemini-1.5 (Reid et al., 2024).

The benchmarks span a variety of domains, including general VQA (SEED-2), knowledge-dense VQA (MMMU), hallucination (POPE), medical question answering (PathVQA), and emotion recognition (FaceEmotion). Some large-scale benchmarks, such as SEED-2 (Li et al., 2023a) and MMMU (Yue et al., 2024), cover multiple tasks. To conduct a fine-grain analysis, we split these benchmarks into task-specific datasets, resulting in 176 datasets in total. Following HEMM (Liang et al., 2024), we subsample some datasets to reduce computational and API calling costs of LVLMs. For each dataset, we calculate a main metric for PMF (either accuracy or BARTScore), and several other metrics, leading to a total of six metrics for PTF modeling. Full details of datasets and models are provided in the supplementary material (Section A).

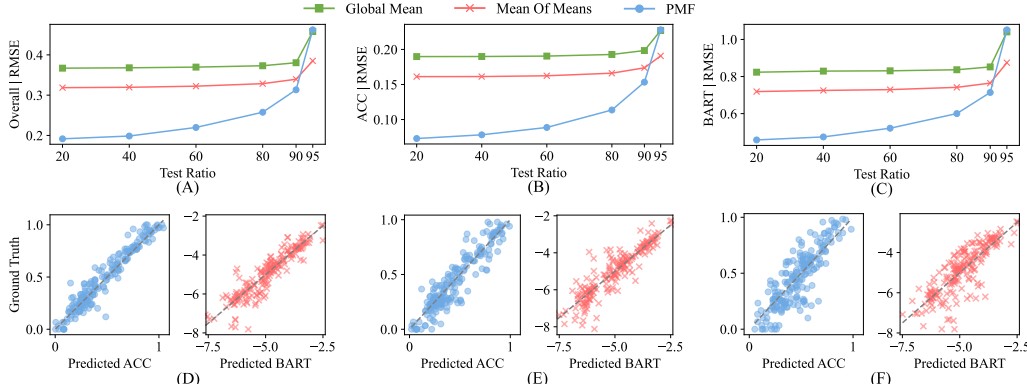

Figure 3: **Performance of PMF.** (A-C) PMF consistently outperforms both baselines when the test ratio is below 90% for estimating all unobserved scores (A), accuracy scores (B), and BART scores (C), with particularly strong performance at lower test ratios. (D-F) The predicted scores exhibit correlations with the ground truth at test ratios of 20% (D), 60% (E), and 90% (F). Gray dashed lines represent perfect prediction i.e., $y = x$. We subsampled 200 scores in (D-F) for visualization.

## 4.2 ESTIMATING UNKNOWN PERFORMANCES

We mask $P\%$ of the elements in the score matrix $\boldsymbol{R}$, use the observed portion to normalize $\boldsymbol{R}$, and train the PMF model using MCMC sampling. The model reconstructs the matrix $\hat{\boldsymbol{R}}$, and we evaluate the performance by comparing the estimated values with the ground truth for the masked elements. For MCMC, we employ the NUTS sampling method, tuning with 500 samples in the burn-in stage and drawing 100 samples. Empirical results show that 100 samples are sufficient for stable estimation. The reconstructed matrix $\hat{\boldsymbol{R}}$ is taken as the mean prediction from MCMC.

We use Root Mean Squared Error (RMSE) as the primary metric to evaluate PMF performance. Additional metrics such as Mean Absolute Error (MAE) and the coefficient of determination ($R^2$) are reported in the supplementary material (Section B).

We compare our method against two baselines: (1) Global Mean: predicting the global mean for unobserved scores; (2) Mean of Means: for each unobserved score, we average the mean performance of the model, the mean performance on the dataset, and the global mean.

**Results.** As shown in Fig. 3(A-C), PMF significantly outperforms the baselines when the test ratio is lower than 90%. This suggests that when only a portion of the scores is available, PMF can infer the unobserved scores with high accuracy. Additionally, as demonstrated in Fig. 3(D-F), the estimated scores strongly correlate with the actual scores.

However, as the amount of observed data decreases, PMF's performance declines as can be expected. In extreme cases where the test ratio exceeds 90%, with limited information about model or dataset performance, PMF can perform worse than predicting the means. We will address this issue in the following sections with our proposed enhancement techniques.

## 4.3 ACTIVE EVALUATION FOR LVLMS

We compare our uncertainty-based approach against two baselines: (1) Random selection of model-dataset pairs, and (2) an oracle approach that selects the pairs with the highest actual errors. In the experiment, we start by masking 80% performance data in the performance matrix. Then, we progressively conduct more LVLM evaluations using three different strategies, and calculate the improvement in performance prediction of PMF with the updated observed data. The experiment is repeated with 10 different random seeds, and we report the averaged improvement.

**Results.** As shown in Fig. 4, our uncertainty-aware method consistently outperforms the random baseline for a fixed budget of evaluations, especially when the amount of extra data is lower than 30%. However, there remains a gap between our method and the oracle approach.

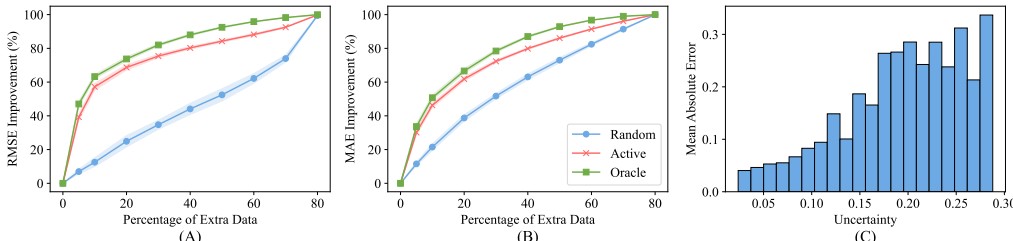

Figure 4: **Comparison of Active Evaluation Methods.** Starting with 20% of the data observed, we progressively conduct additional LVLM evaluations using three different strategies. (A) RMSE and (B) MAE improvement demonstrate the advantage of our method compared to random evaluation. (C) Uncertainties from MCMC are correlated with the actual absolute errors.

Table 1: **Comparison of PMF and PTF.** Superior results are highlighted. PMF (Sep) models each score separately, while PMF (OneMat) combines accuracy and BART scores into a single matrix, as each dataset contains either accuracy or BART scores. PTF is the enhanced model that supports multiple scoring metrics, which outperforms PMF at a high test ratio.

| Method | Overall | | Acc | | Precision | | Recall | | F1 | | BART | | BERT | |
|---|---|---|---|---|---|---|---|---|---|---|---|---|---|---|
| | RMSE↓ | MAE↓ | RMSE | MAE | RMSE | MAE | RMSE | MAE | RMSE | MAE | RMSE | MAE | RMSE | MAE |
| Test Ratio: 20% | | | | | | | | | | | | | | |
| PMF (Sep) | 0.175 | 0.086 | 0.073 | 0.051 | 0.135 | 0.086 | 0.166 | 0.115 | 0.134 | 0.087 | 0.463 | 0.318 | 0.068 | 0.031 |
| PMF (OneMat) | 0.193 | 0.090 | 0.074 | 0.052 | - | - | - | - | - | - | 0.461 | 0.303 | - | - |
| PTF | 0.205 | 0.096 | 0.078 | 0.055 | 0.129 | 0.085 | 0.176 | 0.126 | 0.108 | 0.070 | 0.563 | 0.378 | 0.077 | 0.039 |
| Test Ratio: 90% | | | | | | | | | | | | | | |
| PMF (Sep) | 0.327 | 0.177 | 0.159 | 0.118 | 0.238 | 0.174 | 0.262 | 0.197 | 0.227 | 0.167 | 0.864 | 0.628 | 0.096 | 0.047 |
| PMF (OneMat) | 0.317 | 0.174 | 0.156 | 0.115 | - | - | - | - | - | - | 0.723 | 0.504 | - | - |
| PTF | 0.290 | 0.158 | 0.159 | 0.118 | 0.186 | 0.129 | 0.230 | 0.167 | 0.180 | 0.124 | 0.754 | 0.529 | 0.094 | 0.045 |

## 4.4 ENHANCING PMF

We apply three enhancement techniques to our PMF model and evaluate their effectiveness across different test ratios. To minimize experimental variance, we perform each experiment 10 times with different random seeds and report the average performance at each test ratio.

**Results.** As seen in Table 1, the multi-score method PTF can get better performance when the matrix is very sparse. When there is enough data, separately modeling PMF with each score works very well and is comparable to PTF. For BART and BERT scores, PMF even outperforms PTF. This is likely because PTF assumes a linear relationship between scores. When this assumption does not hold, such as in the case of BART and BERT scores, it can negatively impact model performance. When the test ratio is high, PTF demonstrates better performance.

Fig. 5 illustrates the impact of the other two enhancement techniques. As shown, Bayesian PTF offers only negligible improvements over standard PTF when there is enough observed data, but it is particularly beneficial in sparse conditions. In Fig. 5(B), our custom profiles also show improvements when data is limited, though there remains a gap between our custom profiles and the oracle profiles. Additionally, Fig. 5(C) highlights that adding profiles not only enhances PTF's overall performance but also reduces instability, as seen by smaller error bars. Model profiles show significant performance gains, whereas dataset profiles contribute only marginally. Better methods for encoding and utilizing dataset information need further exploration.

## 5 DISCUSSION

### 5.1 LOW-RANK PROPERTIES OF THE PERFORMANCE MATRIX

We investigate the impact of different latent dimensions in the PMF models and find that a relatively small latent dimension, around 10, is sufficient. As shown in Fig. 6, increasing the latent dimension reduces the RMSE on the training data to zero due to overfitting, but it does not lead to significant

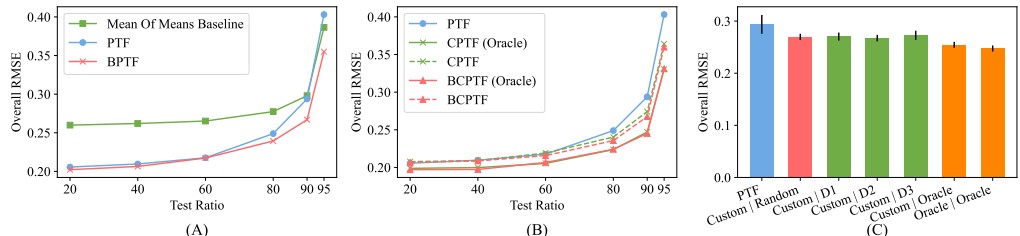

Figure 5: **Performance of Enhanced PTF.** (A) BPTF shows minimal improvement over standard PTF when data is sufficient but proves particularly beneficial under sparse conditions. (B) Custom profiles improve performance when data is limited, though a gap remains compared to oracle profiles. (C) Ablation study on model and dataset profiles. "A | B" represents using A for the model profile and B for the dataset profile. Custom model profiles lead to significant performance gains, while dataset profiles contribute only marginally. BPTF, Bayesian PTF; CPTF, Constrained PTF.

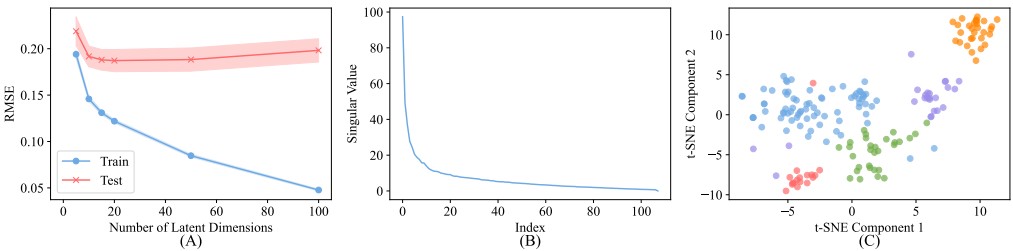

Figure 6: **Low-Rank Property of the Score Matrix.** (A) RMSE on the test set for PMF stabilizes when the latent dimension exceeds 15. (B) The top singular values of the performance matrix are significantly larger than the others. (C) t-SNE visualization of dataset clusters.

improvements in RMSE on the testing data. Additionally, when we extract the singular values of the score matrix, we observe that the top singular values are much larger than the rest, indicating that most of the information is captured by a few dimensions. This suggests a high degree of similarity in performance scores across benchmarks. A detailed correlation analysis of these performance scores is provided in the supplementary material (Section A).

## 5.2 WHAT CAN WE TELL BASED ON VISION ENCODERS?

The Constrained PMF model can capture the impact of model and dataset profiles. Here, we present a showcase analysis focusing on the vision encoder type from the model profiles. Specifically, we calculate the dot product between the feature vector of the vision encoder type, $H_m$, and the feature vector of the dataset, $V_n'$. The calculation result measures the influence of a vision encoder on a task. As shown in Fig. 7, DINO shows improvements on a few datasets compared to CLIP, while FNet, SigLIP, and ViT are less effective in comparison.

## 5.3 WHICH MODELS OR BENCHMARKS ARE MOST INFORMATIVE?

We assess how representative a model is and how informative a benchmark is, by measuring the RMSE improvements of PMF when we add the full results of a model or dataset. The most informative models and tasks are shown in Fig. 8. As observed, strong models like GPT-4, Gemini, and InterLM are more representative than weaker models. This is likely because their performance tends to deviate from the average and, being more general, they reliably reflect the difficulty level of various datasets. Interestingly, the text-to-image generation task is particularly informative. In this task, models must select the correct generated image from four candidates, and we observe that strong models, such as GPT-4, perform significantly better than others. This performance gap leads to larger errors in PMF, so including this dataset can significantly improve the PMF model.

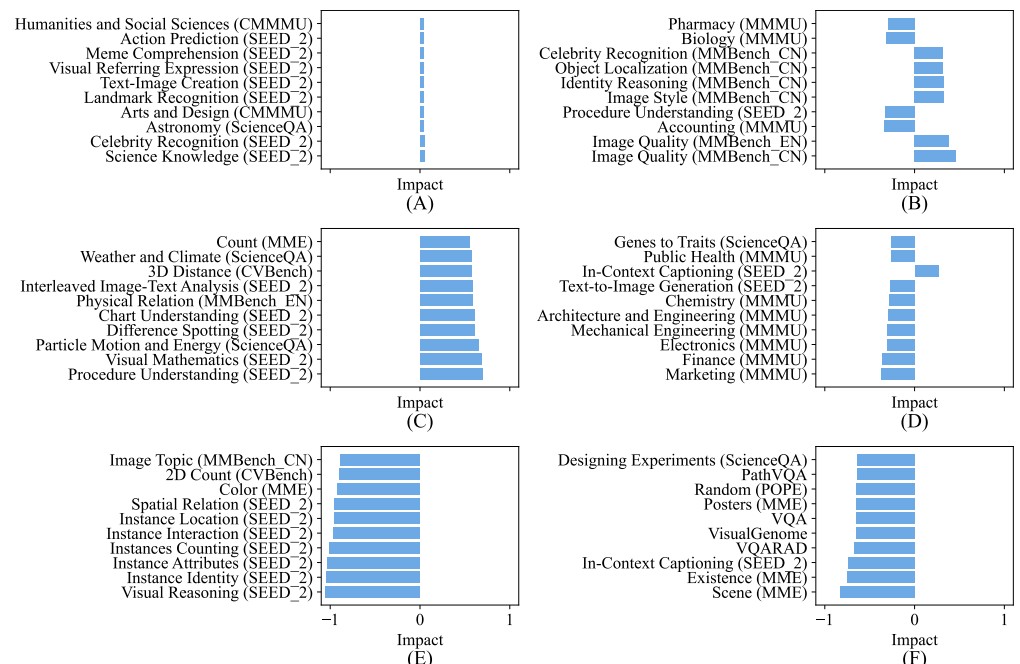

Figure 7: **Effect Analysis of Vision Encoders on Downstream Tasks.** We evaluate the impact of each vision encoder on downstream tasks by calculating the dot product between the feature vector of the vision encoder and the feature vector of the dataset.

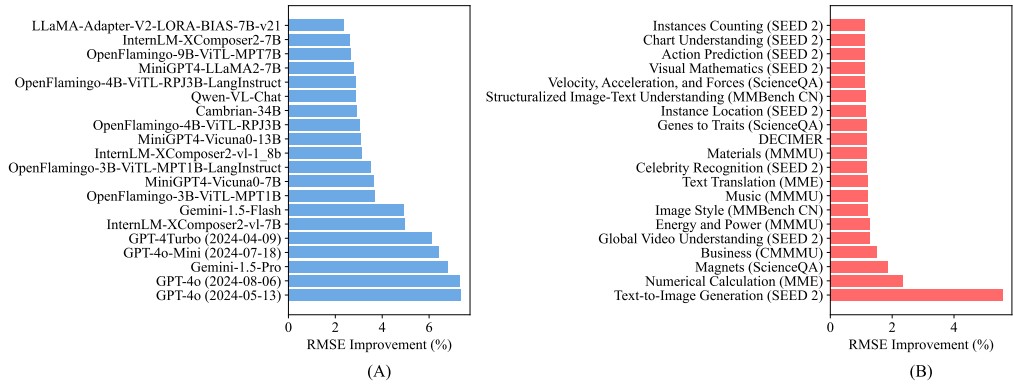

Figure 8: **Which Models and Datasets Are Informative for Performance Estimation.** Given a PMF model train on 20% data of the performance matrix, We measure the improvement in RMSE of PMF when adding the entire results of a model (A) or a dataset (B).

## 6 CONCLUSION AND FUTURE WORK

In this study, we evaluate 108 models on 176 datasets across 36 benchmarks. Our framework estimates unknown LVLM performances across tasks using PMF, prioritizes evaluations based on uncertainty, and introduce some enhancements to address the sparse data issue. Our study could lead to significant savings in development time and computation costs. We highlight several limitations. First, recent advances show that in-context learning or generating multiple responses can improve LVLM performance on the same dataset. Modeling these different evaluation settings (e.g., 5-shot) could extend our framework. Second, some model-dataset pairs with high uncertainties might offer limited value for improving performance prediction on other datasets, so better heuristics for active evaluation could be developed. Third, our method cannot answer what new benchmarks are needed, which we believe is an interesting future direction.

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

## A    COMPREHENSIVE EVALUATION OF LVLMS

We provide a comprehensive overview of the datasets and LVLMs used in our study. Detailed dataset information can be found in Table 2 and 3, while the model profiles are presented in Tables 4 and 5.

A heatmap illustrating the model ranking across datasets is shown in Fig. 9. Additionally, the correlation analysis of performance scores is illustrated in Fig. 10 and 11. Notably, even within the same model family, such as the LLaVA series, the rankings between models do not exhibit a strong correlation. Datasets tend to have much more consistent ranking correlations, suggesting that models performing well on one dataset are likely to rank highly on others as well.

## B    FURTHER EXPERIMENTAL RESULTS

We present detailed performance evaluations of PMF in Table 6 and PTF in Table 7. As shown, our methods consistently outperform the baselines. In scenarios where performance data is sparse, our enhancements significantly improves the prediction accuracy of PMF.

We also investigate the models' ability to generalize to new models and datasets without any performance scores for training. As illustrated in Fig. 12, using model and dataset profiles provides slight improvement for new models or datasets. However, when both the model and dataset are entirely new, performance falls below the Global Mean baseline. But we argue that this situation is rare in practice. Some initial performance scores are usually available when a model or dataset is released, and the community usually reports more performance scores in subsequent works.

Table 2: **Dataset information.** Our study utilizes 36 benchmarks. For larger benchmarks such as SEED-2, we divide them into sub-datasets based on task categories. To reduce computational costs, we subsample certain benchmarks. Download URLs for all benchmarks are provided.

| Benchmark | No. of Datasets | No. of Samples for GPT and Gemini | No. of Samples for Other Models | Download URL |
|---|---|---|---|---|
| SEED 2 (Li et al., 2023a) | 27 | 2606 | 24371 | https://huggingface.co/datasets/lmms-lab/SEED-Bench-2 |
| MME (Fu et al., 2023) | 14 | 1000 | 2374 | https://huggingface.co/datasets/lmms-lab/MME |
| MMBench CN (Liu et al., 2023b) | 20 | 1994 | 4329 | https://huggingface.co/datasets/lmms-lab/MMBench |
| MMBench EN (Liu et al., 2023b) | 20 | 1994 | 4329 | https://huggingface.co/datasets/lmms-lab/MMBench |
| MMMU (Yue et al., 2024) | 30 | 900 | 900 | https://huggingface.co/datasets/lmms-lab/MMMU |
| CMMMU (Zhang et al., 2024a) | 6 | 573 | 900 | https://huggingface.co/datasets/lmms-lab/CMMMU |
| ScienceQA (Lu et al., 2022) | 25 | 1467 | 2017 | https://huggingface.co/datasets/lmms-lab/ScienceQA |
| CVBench (Tong et al., 2024) | 4 | 400 | 2638 | https://huggingface.co/datasets/nyu-visionx/CV-Bench |
| POPE (Li et al., 2023b) | 3 | 900 | 900 | https://github.com/AoiDragon/POPE |
| DECIMER (Brinkhaus et al., 2022) | 1 | 100 | 100 | https://www.kaggle.com/datasets/juliajakubowska/decimer |
| Enrico (Leiva et al., 2020) | 1 | 100 | 100 | https://userinterfaces.aalto.fi/enrico/ |
| FaceEmotion (Goodfellow et al., 2013) | 1 | 100 | 100 | https://www.kaggle.com/datasets/msambare/fer2013 |
| Flickr30k (Young et al., 2014) | 1 | 100 | 100 | https://www.kaggle.com/datasets/hsankesara/flickr-image-dataset |
| GQA (Hudson & Manning, 2019) | 1 | 100 | 100 | https://cs.stanford.edu/people/dorarad/gqa/download.html |
| HatefulMemes (Kiela et al., 2020) | 1 | 100 | 100 | https://www.kaggle.com/datasets/parthplc/facebook-hateful-meme-dataset |
| INAT (Van Horn et al., 2018) | 1 | 100 | 100 | https://ml-inat-competition-datasets.s3.amazonaws.com/2021/val.tar.gz |
| IRFL (Yosef et al., 2023) | 1 | 100 | 100 | https://huggingface.co/datasets/lampent/IRFL |
| MemeCaps (Hwang & Shwartz, 2023) | 1 | 100 | 100 | https://github.com/eujhwang/meme-cap/tree/main |
| Memotion (Sharma et al., 2020) | 1 | 100 | 100 | https://www.kaggle.com/datasets/williamscott701/memotion-dataset-7k |
| MMIMDB (Arevalo et al., 2017) | 1 | 100 | 100 | https://huggingface.co/datasets/akshayg08/mmimdb_test |
| NewYorkerCartoon (Hessel et al., 2022) | 1 | 100 | 100 | https://github.com/nextml/caption-contest-data |
| NLVR (Suhr et al., 2017) | 1 | 100 | 100 | https://github.com/lil-lab/nlvr.git |
| NLVR2 (Suhr et al., 2018) | 1 | 100 | 100 | https://github.com/lil-lab/nlvr.git |
| NoCaps (Agrawal et al., 2019) | 1 | 100 | 100 | https://huggingface.co/datasets/akshayg08/NocapsTest |
| OKVQA (Marino et al., 2019) | 1 | 100 | 100 | https://okvqa.allenai.org/download.html |
| OpenPath (Huang et al., 2023) | 1 | 100 | 100 | https://huggingface.co/datasets/akshayg08/OpenPath |
| PathVQA (He et al., 2020) | 1 | 100 | 100 | https://github.com/UCSD-AI4H/PathVQA |
| Resisc45 (Cheng et al., 2017) | 1 | 100 | 100 | https://www.kaggle.com/datasets/happyyang/nwpu-data-set |
| Screen2Words (Wang et al., 2021) | 1 | 100 | 100 | https://www.kaggle.com/datasets/onurgunes1993/rico-dataset |
| Slake (Liu et al., 2021) | 1 | 100 | 100 | https://huggingface.co/datasets/BoKelvin/SLAKE/ |
| UCMerced (Yang & Newsam, 2010) | 1 | 100 | 100 | https://www.kaggle.com/code/apollo2506/land-scene-classification |
| VCR (Zellers et al., 2019) | 1 | 100 | 100 | https://visualcommonsense.com/download/ |
| VisualGenome (Krishna et al., 2017) | 1 | 100 | 100 | https://homes.cs.washington.edu/ ranjay/visualgenome/ |
| VQA (Antol et al., 2015) | 1 | 100 | 100 | https://visualqa.org/vqa_v1_download.html |
| VQARAD (Lau et al., 2018) | 1 | 100 | 100 | https://huggingface.co/datasets/flaviagiammarino/vqa-rad |
| Winoground (Thrush et al., 2022) | 1 | 100 | 100 | https://huggingface.co/datasets/facebook/winoground |

Table 3: **Dataset Metrics.** PMF models the main metric on the datasets, while PTF utilizes the main and other metrics (six in total) in modeling. BARTScore is proposed by Yuan et al. (2021), while BERTScore is introduced by Zhang et al. (2019).

| Benchmark | Main Metric | Other Metrics |
|---|---|---|
| SEED 2 (Li et al., 2023a) | Accuracy | - |
| MME (Fu et al., 2023) | Accuracy | Precision, Recall, F1 |
| MMBench CN (Liu et al., 2023b) | Accuracy | - |
| MMBench EN (Liu et al., 2023b) | Accuracy | - |
| MMMU (Yue et al., 2024) | Accuracy | - |
| CMMMU (Zhang et al., 2024a) | Accuracy | - |
| ScienceQA (Lu et al., 2022) | Accuracy | - |
| CVBench (Tong et al., 2024) | Accuracy | - |
| POPE (Li et al., 2023b) | Accuracy | Precision, Recall, F1 |
| DECIMER (Brinkhaus et al., 2022) | BARTScore | BERTScore |
| Enrico (Leiva et al., 2020) | BARTScore | BERTScore |
| FaceEmotion (Goodfellow et al., 2013) | BARTScore | BERTScore |
| Flickr30k (Young et al., 2014) | BARTScore | BERTScore |
| GQA (Hudson & Manning, 2019) | BARTScore | BERTScore |
| HatefulMemes (Kiela et al., 2020) | BARTScore | BERTScore |
| INAT (Van Horn et al., 2018) | BARTScore | BERTScore |
| IRFL (Yosef et al., 2023) | BARTScore | BERTScore |
| MemeCaps (Hwang & Shwartz, 2023) | BARTScore | BERTScore |
| Memotion (Sharma et al., 2020) | BARTScore | BERTScore |
| MMIMDB (Arevalo et al., 2017) | BARTScore | BERTScore |
| NewYorkerCartoon (Hessel et al., 2022) | BARTScore | BERTScore |
| NLVR (Suhr et al., 2017) | BARTScore | BERTScore |
| NLVR2 (Suhr et al., 2018) | BARTScore | BERTScore |
| NoCaps (Agrawal et al., 2019) | BARTScore | BERTScore |
| OKVQA (Marino et al., 2019) | BARTScore | BERTScore |
| OpenPath (Huang et al., 2023) | BARTScore | BERTScore |
| PathVQA (He et al., 2020) | BARTScore | BERTScore |
| Resisc45 (Cheng et al., 2017) | BARTScore | BERTScore |
| Screen2Words (Wang et al., 2021) | BARTScore | BERTScore |
| Slake (Liu et al., 2021) | BARTScore | BERTScore |
| UCMerced (Yang & Newsam, 2010) | BARTScore | BERTScore |
| VCR (Zellers et al., 2019) | BARTScore | BERTScore |
| VisualGenome (Krishna et al., 2017) | BARTScore | BERTScore |
| VQA (Antol et al., 2015) | BARTScore | BERTScore |
| VQARAD (Lau et al., 2018) | BARTScore | BERTScore |
| Winoground (Thrush et al., 2022) | BARTScore | BERTScore |

Table 4: **Model Information.** Our study evaluates 108 models. For each model, we report the number of parameters in the LLM backbone, the vision encoder, and the model family that we define.

| Model | Checkpoint | No. Param. in LLM | Vision Encoder | Model Family |
|---|---|---|---|---|
| BLIP2 | BLIP2-opt-2.7B | 2.7 | ViT | BLIP |
| | BLIP2-flan-t5-xxl | 11 | ViT | BLIP |
| | BLIP2-opt-6.7b-coco | 6.7 | ViT | BLIP |
| | BLIP2-opt-6.7b | 6.7 | ViT | BLIP |
| | BLIP2-flan-t5-xl | 3 | ViT | BLIP |
| InstructBLIP | InstructBLIP-Vicuna-7B | 7 | ViT | BLIP |
| | InstructBLIP-Vicuna-13B | 13 | ViT | BLIP |
| | InstructBLIP-flan-t5-xl | 3 | ViT | BLIP |
| | InstructBLIP-flan-t5-xxl | 11 | ViT | BLIP |
| MiniGPT4 | MiniGPT4-LLaMA2-7B | 7 | ViT | MiniGPT4 |
| | MiniGPT4-Vicuna0-7B | 7 | ViT | MiniGPT4 |
| | MiniGPT4-Vicuna0-13B | 13 | ViT | MiniGPT4 |
| mPLUG-Owl | mPLUG-Owl2-LLaMA2-7B | 7 | ViT | MiniGPT4 |
| | mPLUG-Owl2_1 | 7 | ViT | mPLUG-Owl |
| LLaVA | LLaVA-7B | 7 | CLIP | LLaVA |
| | LLaVA-13B | 13 | CLIP | LLaVA |
| | LLaVA-v1.6-Vicuna-7B | 7 | CLIP | LLaVA |
| | LLaVA-v1.6-Vicuna-13B | 13 | CLIP | LLaVA |
| | LLaVA-v1.6-Mistral-7B | 7 | CLIP | LLaVA |
| | LLaVA-v1.6-34B | 34 | CLIP | LLaVA |
| Cambrian-1 | Cambrian-Phi3-3B | 3 | CLIP, SigLIP, ConvNeXt, DINOv2 | Cambrian |
| | Cambrian-8B | 8 | CLIP, SigLIP, ConvNeXt, DINOv2 | Cambrian |
| | Cambrian-13B | 13 | CLIP, SigLIP, ConvNeXt, DINOv2 | Cambrian |
| | Cambrian-34B | 34 | CLIP, SigLIP, ConvNeXt, DINOv2 | Cambrian |
| Fuyu | Fuyu-8B | 8 | | Fuyu |
| LLaMA_Adapter | LLaMA-Adapter-V2-BIAS-7B | 7 | CLIP | LLaMA-Adapter |
| | LLaMA-Adapter-V2-LORA-BIAS-7B | 7 | CLIP | LLaMA-Adapter |
| | LLaMA-Adapter-V2-LORA-BIAS-7B-v21 | 7 | CLIP | LLaMA-Adapter |
| OpenFlamingo | OpenFlamingo-3B-vitl-mpt1b | 1 | NFNet | OpenFlamingo |
| | OpenFlamingo-3B-vitl-mpt1b-langinstruct | 1 | NFNet | OpenFlamingo |
| | OpenFlamingo-4B-vitl-rpj3b | 3 | NFNet | OpenFlamingo |
| | OpenFlamingo-4B-vitl-rpj3b-langinstruct | 3 | NFNet | OpenFlamingo |
| | OpenFlamingo-9B-vitl-mpt7b | 7 | NFNet | OpenFlamingo |
| Qwen-VL | Qwen-VL-Chat | 7 | ViT | Qwen |
| InternLM_XComposer | InternLM-XComposer-7B | 7 | CLIP | InternLM |
| | InternLM-XComposer-vl-7B | 7 | CLIP | InternLM |
| | InternLM-XComposer2-7B | 7 | CLIP | InternLM |
| | InternLM-XComposer2-vl-1_8b | 1.8 | CLIP | InternLM |
| | InternLM-XComposer2-vl-7B | 7 | CLIP | InternLM |
| GPT4 | gpt-4o-2024-05-13 | Unknown | Unknown | GPT4 |
| | gpt-4o-2024-08-06 | Unknown | Unknown | GPT4 |
| | gpt-4o-mini-2024-07-18 | Unknown | Unknown | GPT4 |
| | gpt-4-turbo-2024-04-09 | Unknown | Unknown | GPT4 |
| Gemini | gemini-1.5-pro | Unknown | Unknown | Gemini |
| | gemini-1.5-flash | Unknown | Unknown | Gemini |

Table 5: **Model information.** This is the continued table of Table 4

| Model | Checkpoint | No. Param. in LLM | Vision Encoder | Model Family |
|---|---|---|---|---|
| Prismatic | reproduction-llava-v15+7b | 7 | CLIP | prism |
| | reproduction-llava-v15+13b | 13 | CLIP | prism |
| | one-stage+7b | 7 | CLIP | prism |
| | one-stage+13b | 13 | CLIP | prism |
| | full-ft-multi-stage+7b | 7 | CLIP | prism |
| | full-ft-one-stage+7b | 7 | CLIP | prism |
| | in1k-224px+7b | 7 | ViT | prism |
| | dinov2-224px+7b | 7 | DINOv2 | prism |
| | clip-224px+7b | 7 | CLIP | prism |
| | siglip-224px+7b | 7 | SigLIP | prism |
| | clip-336px-resize-crop+7b | 7 | CLIP | prism |
| | clip-336px-resize-naive+7b | 7 | CLIP | prism |
| | siglip-384px-letterbox+7b | 7 | SigLIP | prism |
| | siglip-384px-resize-crop+7b | 7 | SigLIP | prism |
| | siglip-384px-resize-naive+7b | 7 | SigLIP | prism |
| | dinoclip-336px-letterbox+7b | 7 | CLIP, DINOv2 | prism |
| | dinoclip-336px-resize-naive+7b | 7 | CLIP, DINOv2 | prism |
| | dinosiglip-384px-letterbox+7b | 7 | SigLIP, DINOv2 | prism |
| | dinosiglip-384px-resize-naive+7b | 7 | SigLIP, DINOv2 | prism |
| | llama2+7b | 7 | CLIP | prism |
| | llama2+13b | 13 | CLIP | prism |
| | vicuna-no-cotraining+7b | 7 | CLIP | prism |
| | llama2-no-cotraining+7b | 7 | CLIP | prism |
| | train-1.25-epochs+7b | 7 | CLIP | prism |
| | train-1.5-epochs+7b | 7 | CLIP | prism |
| | train-2-epochs+7b | 7 | CLIP | prism |
| | train-3-epochs+7b | 7 | CLIP | prism |
| | llava-lvis4v+7b | 7 | CLIP | prism |
| | llava-lrv+7b | 7 | CLIP | prism |
| | llava-lvis4v-lrv+7b | 7 | CLIP | prism |
| | prism-clip-controlled+7b | 7 | CLIP | prism |
| | prism-clip-controlled+13b | 13 | CLIP | prism |
| | prism-clip+7b | 7 | CLIP | prism |
| | prism-clip+13b | 13 | CLIP | prism |
| | prism-siglip-controlled+7b | 7 | SigLIP | prism |
| | prism-siglip-controlled+13b | 13 | SigLIP | prism |
| | prism-siglip+7b | 7 | SigLIP | prism |
| | prism-siglip+13b | 13 | SigLIP | prism |
| | prism-dinosiglip-controlled+7b | 7 | SigLIP, DINOv2 | prism |
| | prism-dinosiglip-controlled+13b | 13 | SigLIP, DINOv2 | prism |
| | prism-dinosiglip+7b | 7 | SigLIP, DINOv2 | prism |
| | prism-dinosiglip+13b | 13 | SigLIP, DINOv2 | prism |
| | prism-dinosiglip-224px-controlled+7b | 7 | SigLIP, DINOv2 | prism |
| | prism-dinosiglip-224px+7b | 7 | SigLIP, DINOv2 | prism |
| | llama2-chat+13b | 13 | CLIP | prism |
| | mistral-v0.1+7b | 7 | CLIP | prism |
| | mistral-instruct-v0.1+7b | 7 | CLIP | prism |
| | phi-2+3b | 3 | CLIP | prism |
| | gemma-instruct+2b+clip | 2 | CLIP | prism |
| | gemma-instruct+2b+siglip | 2 | SigLIP | prism |
| | gemma-instruct+2b+dinosiglip | 2 | SigLIP, DINOv2 | prism |
| | gemma-instruct+8b+clip | 8 | CLIP | prism |
| | gemma-instruct+8b+siglip | 8 | SigLIP | prism |
| | gemma-instruct+8b+dinosiglip | 8 | SigLIP, DINOv2 | prism |
| | llama2-chat+7b+clip | 7 | CLIP | prism |
| | llama2-chat+7b+siglip | 7 | SigLIP | prism |
| | llama2-chat+7b+dinosiglip | 7 | SigLIP, DINOv2 | prism |
| | llama3-instruct+8b+clip | 8 | CLIP | prism |
| | llama3-instruct+8b+siglip | 8 | SigLIP | prism |
| | llama3-instruct+8b+dinosiglip | 8 | SigLIP, DINOv2 | prism |
| | mistral-instruct-v0.2+7b+clip | 7 | CLIP | prism |
| | mistral-instruct-v0.2+7b+siglip | 7 | SigLIP | prism |
| | mistral-instruct-v0.2+7b+dinosiglip | 7 | SigLIP, DINOv2 | prism |

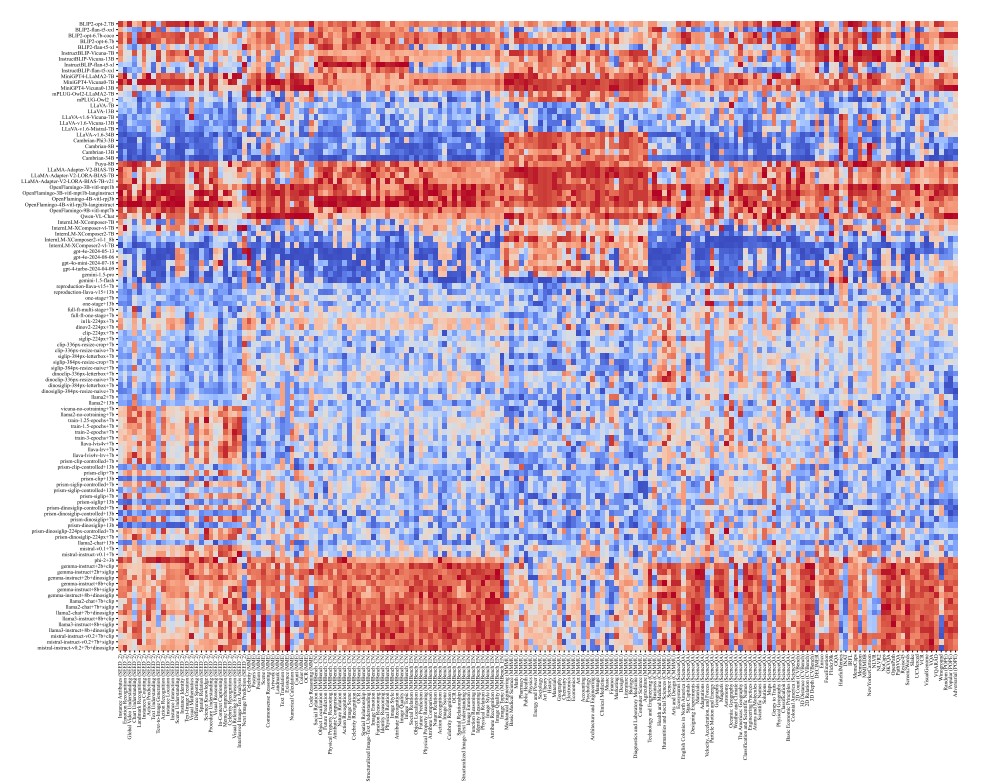

Figure 9: **Heatmap of Model Rankings on Each Dataset.**

Table 6: **Detailed performance of PMF.** Superior results are highlighted.

| Method | Overall | | Acc | | | BART | | |
|---|---|---|---|---|---|---|---|---|
| | RMSE↓ | MAE↓ | RMSE | MAE | $R^2$ ↑ | RMSE | MAE | $R^2$ |
| Test Ratio: 20% | | | | | | | | |
| Global Mean | 0.367 | 0.220 | 0.190 | 0.148 | 0.475 | 0.823 | 0.611 | 0.549 |
| Mean Of Means | 0.319 | 0.186 | 0.161 | 0.125 | 0.622 | 0.719 | 0.525 | 0.656 |
| PMF | 0.192 | 0.090 | 0.073 | 0.051 | 0.922 | 0.458 | 0.303 | 0.860 |
| Test Ratio: 40% | | | | | | | | |
| Global Mean | 0.368 | 0.220 | 0.190 | 0.149 | 0.477 | 0.829 | 0.614 | 0.551 |
| Mean Of Means | 0.320 | 0.186 | 0.161 | 0.125 | 0.623 | 0.725 | 0.527 | 0.657 |
| PMF | 0.199 | 0.095 | 0.078 | 0.056 | 0.911 | 0.474 | 0.314 | 0.853 |
| Test Ratio: 60% | | | | | | | | |
| Global Mean | 0.370 | 0.220 | 0.190 | 0.149 | 0.474 | 0.831 | 0.613 | 0.551 |
| Mean Of Means | 0.322 | 0.188 | 0.162 | 0.126 | 0.618 | 0.729 | 0.529 | 0.654 |
| PMF | 0.220 | 0.106 | 0.089 | 0.063 | 0.886 | 0.521 | 0.348 | 0.823 |
| Test Ratio: 80% | | | | | | | | |
| Global Mean | 0.373 | 0.221 | 0.193 | 0.150 | 0.462 | 0.837 | 0.612 | 0.546 |
| Mean Of Means | 0.329 | 0.191 | 0.166 | 0.128 | 0.601 | 0.742 | 0.533 | 0.643 |
| PMF | 0.258 | 0.131 | 0.114 | 0.081 | 0.812 | 0.600 | 0.407 | 0.766 |
| Test Ratio: 90% | | | | | | | | |
| Global Mean | 0.381 | 0.226 | 0.198 | 0.153 | 0.430 | 0.852 | 0.630 | 0.529 |
| Mean Of Means | 0.339 | 0.197 | 0.174 | 0.133 | 0.564 | 0.765 | 0.555 | 0.621 |
| PMF | 0.313 | 0.172 | 0.153 | 0.113 | 0.660 | 0.714 | 0.502 | 0.669 |
| Test Ratio: 95% | | | | | | | | |
| Global Mean | 0.458 | 0.278 | 0.227 | 0.182 | 0.254 | 1.041 | 0.807 | 0.297 |
| Mean Of Means | 0.385 | 0.230 | 0.191 | 0.150 | 0.474 | 0.875 | 0.672 | 0.504 |
| PMF | 0.462 | 0.276 | 0.228 | 0.180 | 0.248 | 1.052 | 0.805 | 0.282 |

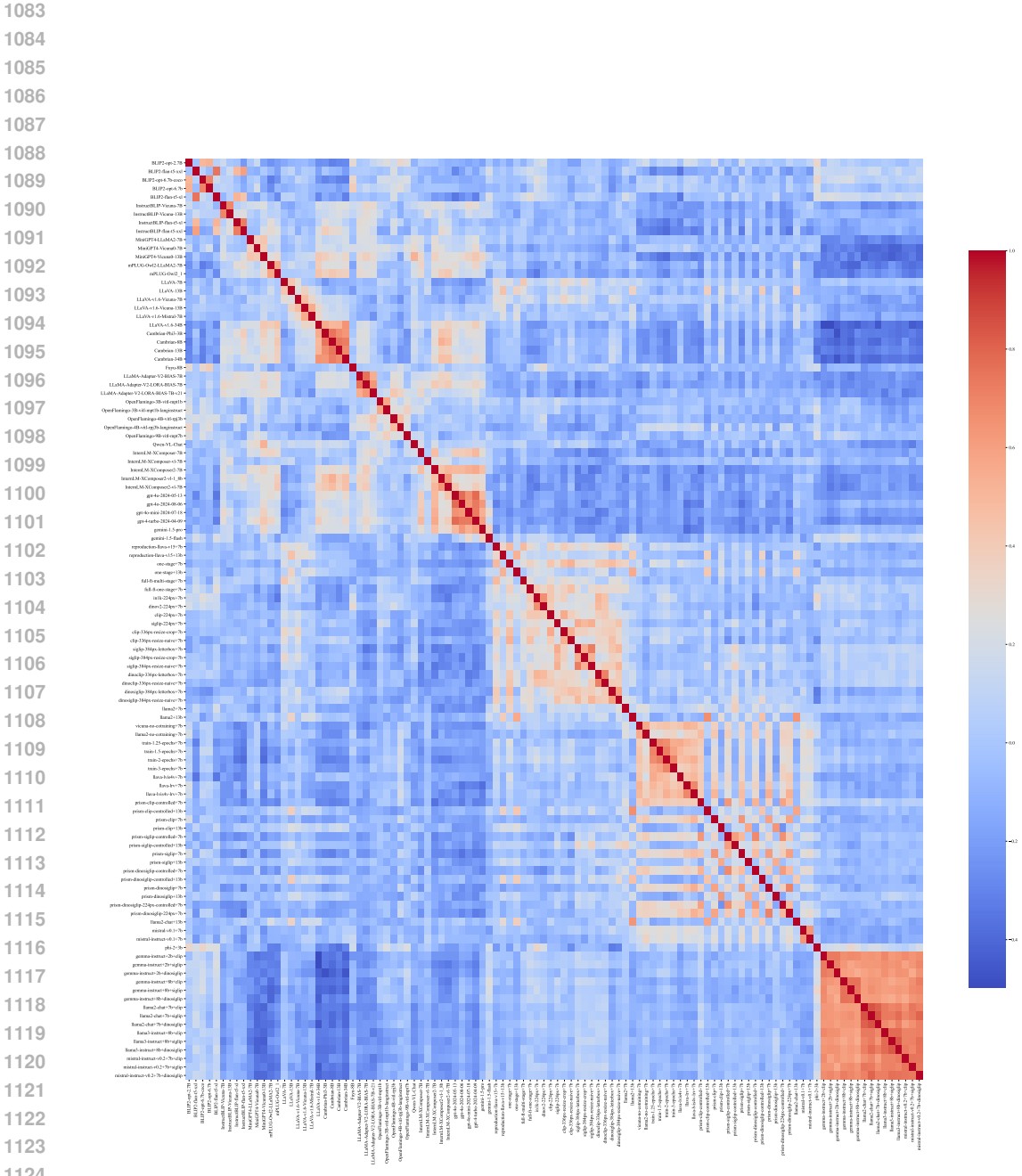

Figure 10: **Heatmap of Correlation in Model Ranking.**

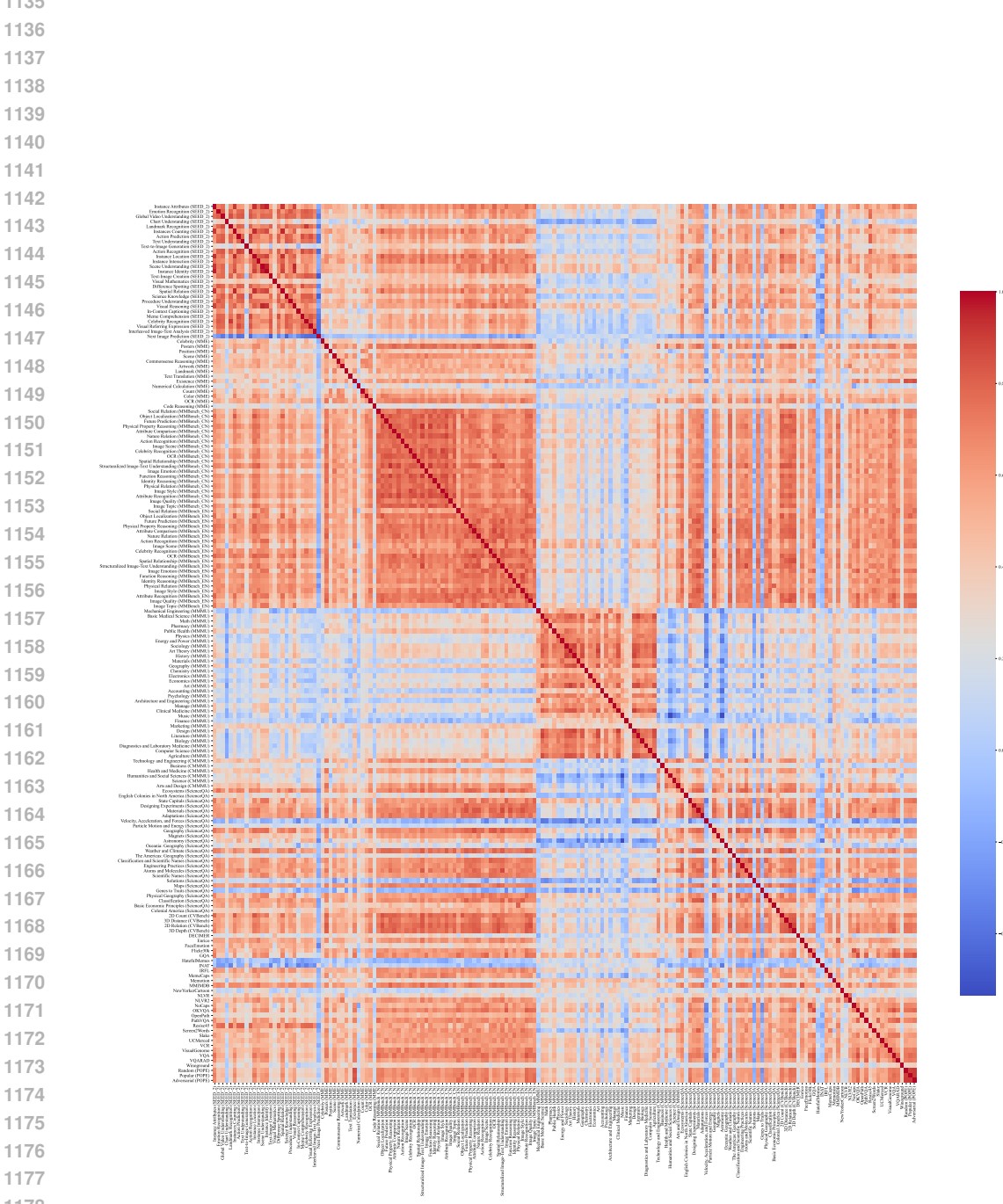

Figure 11: **Heatmap of Ranking Correlation on Datasets.**

Table 7: **Detailed performance of PTF.** Superior results are highlighted.

| Method | Overall RMSE↓ | Overall MAE↓ | Acc RMSE | Acc MAE | Precision RMSE | Precision MAE | Recall RMSE | Recall MAE | F1 RMSE | F1 MAE | BART RMSE | BART MAE | BERT RMSE | BERT MAE |
|---|---|---|---|---|---|---|---|---|---|---|---|---|---|---|
| **Test Ratio: 20%** | | | | | | | | | | | | | | |
| Global Mean | 0.320 | 0.190 | 0.190 | 0.149 | 0.223 | 0.167 | 0.245 | 0.186 | 0.205 | 0.156 | 0.812 | 0.603 | 0.088 | 0.044 |
| Mean Of Means | 0.260 | 0.150 | 0.149 | 0.115 | 0.181 | 0.131 | 0.205 | 0.152 | 0.169 | 0.123 | 0.664 | 0.482 | 0.074 | 0.036 |
| PMF | 0.206 | 0.096 | 0.081 | 0.057 | 0.130 | 0.086 | 0.175 | 0.126 | 0.108 | 0.070 | 0.563 | 0.378 | 0.077 | 0.039 |
| CPTF | 0.208 | 0.099 | 0.087 | 0.060 | 0.134 | 0.089 | 0.173 | 0.123 | 0.107 | 0.070 | 0.564 | 0.379 | 0.076 | 0.039 |
| BPTF | 0.202 | 0.095 | 0.079 | 0.056 | 0.129 | 0.084 | 0.177 | 0.127 | 0.109 | 0.070 | 0.553 | 0.372 | 0.077 | 0.039 |
| BCPTF | 0.207 | 0.096 | 0.079 | 0.056 | 0.129 | 0.085 | 0.178 | 0.127 | 0.113 | 0.072 | 0.568 | 0.378 | 0.076 | 0.039 |
| **Test Ratio: 40%** | | | | | | | | | | | | | | |
| Global Mean | 0.323 | 0.192 | 0.190 | 0.149 | 0.223 | 0.167 | 0.247 | 0.189 | 0.213 | 0.160 | 0.818 | 0.609 | 0.091 | 0.045 |
| Mean Of Means | 0.262 | 0.151 | 0.150 | 0.116 | 0.182 | 0.132 | 0.209 | 0.156 | 0.174 | 0.126 | 0.667 | 0.486 | 0.078 | 0.038 |
| PMF | 0.210 | 0.100 | 0.083 | 0.059 | 0.132 | 0.087 | 0.181 | 0.130 | 0.112 | 0.073 | 0.572 | 0.388 | 0.081 | 0.040 |
| CPTF | 0.209 | 0.101 | 0.087 | 0.062 | 0.134 | 0.089 | 0.180 | 0.128 | 0.113 | 0.074 | 0.566 | 0.385 | 0.081 | 0.040 |
| BPTF | 0.206 | 0.100 | 0.084 | 0.060 | 0.132 | 0.086 | 0.185 | 0.133 | 0.117 | 0.077 | 0.558 | 0.379 | 0.082 | 0.041 |
| BCPTF | 0.209 | 0.100 | 0.082 | 0.059 | 0.131 | 0.086 | 0.184 | 0.131 | 0.117 | 0.076 | 0.568 | 0.384 | 0.081 | 0.040 |
| **Test Ratio: 60%** | | | | | | | | | | | | | | |
| Global Mean | 0.325 | 0.192 | 0.191 | 0.149 | 0.227 | 0.170 | 0.248 | 0.189 | 0.214 | 0.160 | 0.825 | 0.611 | 0.092 | 0.045 |
| Mean Of Means | 0.265 | 0.153 | 0.151 | 0.116 | 0.186 | 0.135 | 0.212 | 0.157 | 0.178 | 0.128 | 0.676 | 0.490 | 0.080 | 0.038 |
| PMF | 0.218 | 0.107 | 0.093 | 0.067 | 0.136 | 0.090 | 0.188 | 0.134 | 0.123 | 0.081 | 0.588 | 0.400 | 0.084 | 0.041 |
| CPTF | 0.219 | 0.109 | 0.098 | 0.070 | 0.141 | 0.094 | 0.187 | 0.133 | 0.125 | 0.082 | 0.588 | 0.398 | 0.083 | 0.040 |
| BPTF | 0.217 | 0.108 | 0.096 | 0.068 | 0.138 | 0.092 | 0.194 | 0.139 | 0.130 | 0.087 | 0.582 | 0.397 | 0.085 | 0.042 |
| BCPTF | 0.216 | 0.105 | 0.089 | 0.064 | 0.135 | 0.090 | 0.191 | 0.136 | 0.127 | 0.083 | 0.584 | 0.394 | 0.083 | 0.041 |
| **Test Ratio: 80%** | | | | | | | | | | | | | | |
| Global Mean | 0.330 | 0.194 | 0.193 | 0.150 | 0.230 | 0.171 | 0.253 | 0.191 | 0.217 | 0.162 | 0.839 | 0.619 | 0.092 | 0.046 |
| Mean Of Means | 0.277 | 0.158 | 0.155 | 0.119 | 0.198 | 0.141 | 0.225 | 0.165 | 0.186 | 0.134 | 0.709 | 0.510 | 0.084 | 0.041 |
| PMF | 0.249 | 0.128 | 0.120 | 0.087 | 0.151 | 0.103 | 0.207 | 0.148 | 0.145 | 0.098 | 0.661 | 0.457 | 0.091 | 0.044 |
| CPTF | 0.240 | 0.123 | 0.115 | 0.083 | 0.151 | 0.104 | 0.208 | 0.148 | 0.147 | 0.099 | 0.637 | 0.437 | 0.088 | 0.043 |
| BPTF | 0.239 | 0.123 | 0.116 | 0.083 | 0.152 | 0.103 | 0.212 | 0.151 | 0.151 | 0.102 | 0.630 | 0.433 | 0.090 | 0.044 |
| BCPTF | 0.236 | 0.119 | 0.108 | 0.077 | 0.147 | 0.099 | 0.208 | 0.149 | 0.147 | 0.099 | 0.627 | 0.427 | 0.089 | 0.043 |
| **Test Ratio: 90%** | | | | | | | | | | | | | | |
| Global Mean | 0.338 | 0.198 | 0.199 | 0.154 | 0.237 | 0.174 | 0.258 | 0.195 | 0.224 | 0.166 | 0.858 | 0.629 | 0.095 | 0.047 |
| Mean Of Means | 0.298 | 0.168 | 0.166 | 0.125 | 0.216 | 0.153 | 0.239 | 0.176 | 0.204 | 0.147 | 0.764 | 0.547 | 0.090 | 0.043 |
| PMF | 0.294 | 0.161 | 0.161 | 0.119 | 0.194 | 0.135 | 0.235 | 0.171 | 0.187 | 0.131 | 0.761 | 0.535 | 0.094 | 0.045 |
| CPTF | 0.274 | 0.147 | 0.145 | 0.105 | 0.190 | 0.133 | 0.233 | 0.168 | 0.184 | 0.128 | 0.710 | 0.492 | 0.092 | 0.043 |
| BPTF | 0.267 | 0.143 | 0.142 | 0.103 | 0.179 | 0.123 | 0.232 | 0.167 | 0.178 | 0.122 | 0.690 | 0.480 | 0.093 | 0.044 |
| BCPTF | 0.268 | 0.141 | 0.138 | 0.099 | 0.179 | 0.124 | 0.228 | 0.164 | 0.176 | 0.120 | 0.698 | 0.481 | 0.093 | 0.045 |
| **Test Ratio: 95%** | | | | | | | | | | | | | | |
| Global Mean | 0.404 | 0.238 | 0.228 | 0.182 | 0.251 | 0.194 | 0.270 | 0.209 | 0.240 | 0.183 | 1.058 | 0.805 | 0.101 | 0.057 |
| Mean Of Means | 0.387 | 0.217 | 0.202 | 0.158 | 0.241 | 0.182 | 0.261 | 0.198 | 0.230 | 0.172 | 1.027 | 0.772 | 0.097 | 0.056 |
| PMF | 0.403 | 0.233 | 0.223 | 0.177 | 0.244 | 0.185 | 0.269 | 0.204 | 0.236 | 0.175 | 1.059 | 0.801 | 0.101 | 0.057 |
| CPTF | 0.364 | 0.199 | 0.188 | 0.142 | 0.216 | 0.161 | 0.252 | 0.188 | 0.216 | 0.157 | 0.970 | 0.712 | 0.097 | 0.054 |
| BPTF | 0.355 | 0.196 | 0.189 | 0.144 | 0.208 | 0.153 | 0.251 | 0.188 | 0.206 | 0.148 | 0.940 | 0.687 | 0.098 | 0.055 |
| BCPTF | 0.360 | 0.196 | 0.186 | 0.141 | 0.211 | 0.156 | 0.251 | 0.186 | 0.205 | 0.148 | 0.959 | 0.702 | 0.100 | 0.056 |

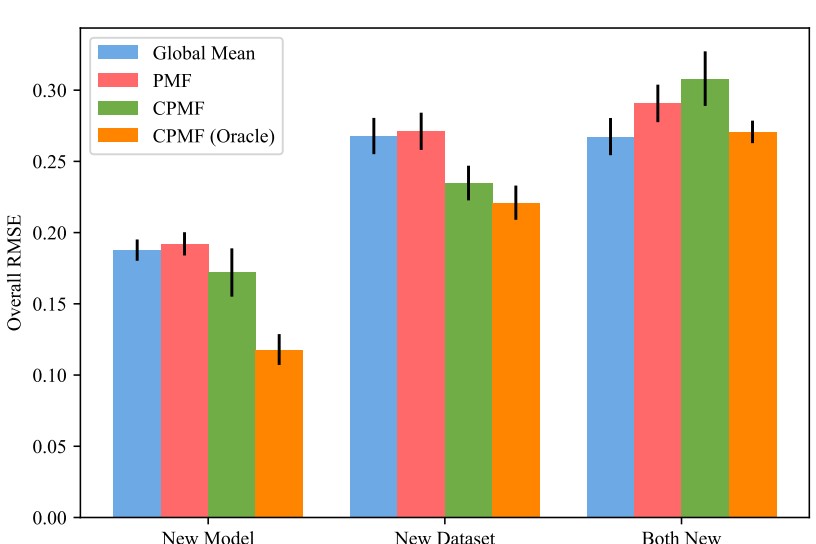

Figure 12: **Results on Purely New Models and Datasets.**

