# OpenReview forum: "Can We Predict Performance of Large Models across Vision-Language Tasks?"
_ICLR.cc/2025/Conference — Submitted to ICLR 2025_

### Official Review · Reviewer_dktp · 2024-10-21

**Soundness:** 2
**Presentation:** 3
**Contribution:** 2
**Rating:** 3
**Confidence:** 3

**Summary:**

This paper presents a framework for predicting unknown performance scores of LVLMs by formulating it as a matrix completion task using probabilistic matrix factorization with MCMC. The paper addresses the challenge of high computational costs in evaluating LVLMs and aims to reduce unnecessary evaluations by predicting performance scores based on observed ones from other models or tasks.

**Strengths:**

1. This paper evaluates 108 models on 176 datasets, covering a wide range of tasks and benchmarks. This systematic evaluation can provide a foundation for many future research.
2. PMF for handling sparse data, such as tensor factorization, Bayesian PMF, and the use of model and dataset profiles, seems a more robust approach to mitigate potential weaknesses in the matrix completion task.

**Weaknesses:**

Of course, it is statistically possible to make more robust predictions, but even humans can predict the performance level of a model to some extent just by observing certain patterns in the results. However, the reasons we still need to directly evaluate are:
1. The learning methodology may show significant weaknesses or strengths on specific benchmarks, and such frameworks cannot analyze these.
2. K-shot, certain promptings, or new evaluation methods could lead to changes in results across benchmarks, but this framework lacks insights into these aspects.

Therefore, although we can statistically predict the results to some extent without directly evaluating a new model, we still confirm the actual performance through evaluation. Moreover, even testing just 10% less in a setting where only a subset of the test set is used can significantly undermine the reliability, making it even harder to trust this framework. In short, using this framework to predict scientific conclusions presents a risk that far outweighs the cost savings.

**Questions:**

A more detailed analysis is needed. Conducting PMF based on different learning methodologies, evaluation pipelines, and promptings could improve the quality of the paper.

---

> ### Author Response · Authors · 2024-11-14
> **Reply - Benefits and Reliability of Our Framework**
>
> We greatly appreciate your time and effort in reviewing our paper. We are pleased that you acknowledged our extensive evaluation and our enhancements to address the sparse-data issue.
>
> > **We still need to directly evaluate.**
>
> We agree with you that direct evaluation is very helpful and important. However, it is expensive to comprehensively evaluate LVLMs. Zhang et al. [3] reports that it takes hundreds of hours to evaluate one model on around 50 tasks in LMMs-Eval, and evaluation even exceeds 1,400 hours on models of 100B parameters or more. With the growth of LVLM benchmarks and models, the evaluation will be more costly.
>
> Our framework can benefit LVLM evaluation in two ways.
>
> * First, it can reduce unnecessary evaluation, given a limited computational budget in practice.
> * Second, it can prioritize the direct evaluation experiments by using our uncertainty-based active evaluation method.
>
> Thus, we respectfully argue that our framework is useful and valuable in practice, which is acknowledged by Reviewers uZXr, VvNb, and Q3Aj.
>
> >  **Reliability of reducing test sets and how to trust our framework.**
>
> We first reference related works to support our paper and then highlight the role of uncertainty estimation in our framework.
>
> Recent works select a coreset of samples from a large benchmark for evaluating LLMs [1, 2] or LVLMs [3, 4]. The performance of a specific model on the coreset is used to estimate its performance on the full benchmark. Thus, it is possible to predict model performance while maintaining reliability.
>
> Moreover, our framework provides uncertainty in performance prediction, which is correlated with the actual absolute errors in Figure 4. The estimated uncertainties can help identify wrong predictions.
>
> We hope that our response can addresses your concern.
>
>
>
> ----
>
> Our references:
>
> Efficient LLM evaluation:
>
> [1] tinyBenchmarks: evaluating LLMs with fewer examples. ICML 2024.
>
> [2] Efficient benchmarking (of language models). NAACL 2024.
>
> Efficient LVLM evaluation:
>
> [3] LMMs-Eval: Reality Check on the Evaluation of Large Multimodal Models. https://arxiv.org/abs/2407.12772.
>
> [4] LIME: Less Is More for MLLM Evaluation. https://arxiv.org/abs/2409.06851.

---

> > ### Comment · Reviewer_dktp · 2024-11-16
> >
> > Thank you for introducing the related pioneering works. Since my position is more negative compared to other reviews, I tried to understand the work in more detail by reading both the suggested lines of work and other reviews.
> >
> > However, I still have questions about the nature of this paper's contribution. The TinyBenchmarking and Efficient Benchmarking papers you introduced are indeed valuable. As pioneers in this field, they proved that we don't need to evaluate LLMs on all test sets, and that evaluating only subsets according to given results can be more reliable and efficient. However, when considering what this paper contributes on top of these existing lines of work, it appears to mainly suggest using already known statistical methods for more accurate predictions. Compared to LLMs, LVLMs have less standardized evaluation pipelines and inference methods, leading to variability and instability in performance across benchmarking. However, from my understanding, this paper seems to have a narrow contribution in the LVLM field - it less analyzes how LVLM variables affect efficient benchmarking or what factors make uncertainty-based approaches more effective as baselines or other novel insights.
> >
> > I truly appreciate the effort put into writing this paper and engaging in discussions with reviewers. If I am interpreting the paper's contribution too narrowly, I apologize, and I remain open to further discussion to understand the paper better.

---

> ### Author Response · Authors · 2024-11-16
>
> We sincerely appreciate your time and effort in reviewing our paper. We summarize your main points and try to address them one by one.
>
>
>
> > **Contributions of our paper, compared to tinyBenchmarks and Efficient Benchmarking.**
>
> We would like to highlight that our framework is different from previous works [1-4].
>
> **Different problem settings.** We predict unknown model performance scores based on known ones across benchmarks and models, while tinyBenchmarks and Efficient benchmarking focus on reducing the size of a benchmark.
>
> For example, let's evaluate models A and B on the SEED and MMMU benchmarks. We are thinking about whether we could use the performance of A on MMMU to predict that of A on SEED, or use the performance of B on MMMU to predict that of A on MMMU. Our goal is to reduce the total number of evaluations. While related works [1-4] aim to reduce the size of SEED and MMMU benchmarks, making each-time evaluation more efficient.
>
> **Different methods.** We formulate our problem as a matrix completion task and use PMF with MCMC to solve it, while previous works usually rely on coreset-selection methods.
>
>
>
> > **Compared to LLMs, LVLMs have less standardized evaluation pipelines and inference methods.**
>
> We are sorry that we are not sure what you mean by "LVLMs have less standardized evaluation pipelines". There are great benchmarks such as SEED, MMBench, MMMU, and POPE. We have included these benchmarks in our experiments. There are also standardized and generalized evaluation pipelines like LMMs-eval and VLMEvalKit. We build our study on these pipelines. We could provide a more specific answer if you would like to provide more explanation.
>
>
>
> >  **Narrow contribution in the LVLM field**
>
> We respectfully highlight that we conduct a comprehensive evaluation of 108 models on 176 datasets, covering a wide range of tasks and benchmarks. This systematic evaluation can provide a foundation for future research.
>
> Moreover, in the Dicussion section of our paper, we conduct further analysis on the correlation of model performances, what are the effects of vision encoders in LVLM on benchmarking, and which LVLMs or benchmark results are more informative to performance prediction.
>
>
>
> If you have any more questions or suggestions, we are happy to discuss them with you.

---

> > ### Comment · Reviewer_dktp · 2024-11-18
> >
> > Before giving my answer, I want to point out that performance varies dramatically depending on several factors, whether for LLMs or LVLMs: how they're evaluated, how they perform reasoning, whether they select the highest probability next token from candidate answers, whether they're prompted to choose from candidates, and whether they're guided toward and evaluated on fine-grained answers. LLMs have developed somewhat standardized pipelines through their relatively long history of extensive research. However, for LVLMs, we're only now beginning to study variables such as cases where they follow correct reasoning paths but give wrong answers when viewing counterfactual images, or where they provide incorrect answers due to language priors despite looking at images. Fine-grained evaluation metrics are also only slowly emerging. These factors represent significant variables in LVLM evaluation unless tested on at least a small test set.
> >
> > 1. Yes, the two pioneering works focus on how to confidently evaluate large-scale test sets with minimal sampling, while this paper addresses predicting benchmark performance without any actual predictions. This is where opinions conflicted, as mentioned in my first comment. Can we really consider the latter problem an advancement over the former? The former is 'evaluation-agnostic' as it finds confidence by testing a few cases and finding an essential subset regardless of any evaluation pipelines, while the latter is not. Especially with LVLMs, where many phenomena remain unanalyzed, making purely statistical predictions in a zero-shot manner is extremely risky. This is what makes me most hesitant to change my position.
> > 2. While perhaps less significant than the above issues, I still don't see the novelty in filling matrix gaps using known statistical techniques. It feels like a step backward from existing active coreset finding methodologies.
> >
> > As mentioned in my first comment, this research's major strength lies in its experimental reporting across numerous models and benchmarks, which could be valuable for various future applications. While I greatly appreciate this aspect, from a 'scientific' perspective, I still find it difficult to change my position.

---

> > > ### Author Response · Authors · 2024-11-18
> > > **Reply 1**
> > >
> > > Thank you for your quick reply and feedback! We summarize your main points and will address them one by one.
> > >
> > > > **LLMs have some standardized evaluation pipelines while LVLMs do not.**
> > >
> > > **Solid foundation for us.** We respectfully disagree with the opinion that "LVLMs have no standardized evaluation pipelines".
> > >
> > > As we listed in the previous reply, there are great works on benchmarking LVLMs, such as
> > >
> > > - SEED-2 (CVPR2024). https://github.com/AILab-CVC/SEED-Bench
> > >
> > > - MMMU (CVPR2024 Award Candidate). https://mmmu-benchmark.github.io/
> > >
> > > - MMBench (ECCV2024 Oral). https://github.com/open-compass/MMBench
> > >
> > > Besides, there are generalized pipelines in LVLMs, such as
> > >
> > > - LMMs-Eval (2.1k stars). https://github.com/EvolvingLMMs-Lab/lmms-eval/tree/main
> > >
> > > - VLMEvalKit (1.3k stars). https://github.com/open-compass/VLMEvalKit/tree/main
> > >
> > > All of them provide codes, leaderboard, and evaluation guidelines for following works to follow. We build our study on these benchmarks and pipelines.
> > >
> > > While model performance under different evaluation settings is different, we respectfully argue that researchers and developers typically follow established benchmarks and pipelines to ensure fair comparisons. This practice helps to significantly reduce variations in evaluation results.
> > >
> > > **Similarities between LLMs and LVLMs.** Moreover, we would like to highlight the similarities between LLMs and LVLMs, such as their architectures, prompting strategies, and decoding techniques.
> > >
> > > LLMs => LVLMs. Although LVLM research has a shorter history than LLM, LVLM evaluation can benefit greatly by adopting successful practices from LLMs.
> > >
> > > LVLMs => LLMs. Our proposed problem and formulation can also be adopted to LLM evaluation.
> > >
> > > Thus, we would like to argue that there are no significant differences between LLMs and LVLMs that reduce the value of our work.
> > >
> > > > **Capture significant variables in LVLM evaluation.**
> > >
> > > If we want to evaluate LVLMs in varying settings, our framework is extensible to different evaluation settings, such as various prompts or decoding strategies.
> > >
> > > - Additional Models. A straightforward way is to treat a model under different evaluation settings as different models, such as "LLaVA (Chain-of-Thought)" and "LLaVA (Beam Search)".
> > > - Additional Profile. In Section 3.5, we introduce model and dataset information for better performance prediction. An extension is to encode evaluation settings as extra information into PMF.
> > >
> > > Specifically, we evaluate LLaVA-v1.5-7B on the 27 tasks in SEED-2, with various evaluation settings. We will test the two methods to extend our framework.
> > >
> > > - Image input. (1) Default: use the clean images, or (2) add Gaussian noise into the images.
> > > - Prompt. (1) Default: prompt the model to choose option ("Answer with the option's letter from the given choices directly."), (2) provide no hint, or (3) use the Chain-of-Thought (CoT) prompt ("Let's think step by step, and then answer with the option's letter.").
> > > - Model decoding. (1) Default: greedy decoding, (2) sampling with temperature = 0.2, (3) sampling with temperature = 0.5, or (4) beam search with temperature = 0.2 and the number of beams = 10.
> > >
> > > We add the results under different evaluation settings into our framework and simply use PMF for prediction.
> > >
> > > The following table reports RMSE of different methods and indicate that our framework can handle different evaluation settings.
> > >
> > > | Method            | Overall   | Default   | Gaussian Noise | No Hint   | CoT       | Sampling (t=0.2) | Sampling (t=0.5) | Beam Search |
> > > | ---- | --------- | --------- | --- | --------- | --------- | ---- | --------- | ----------- |
> > > | *Test Ratio: 20%* |           |           | |           |           |          |        |    |
> > > | Global Mean       | 0.119     | 0.112     | 0.105          | 0.090     | 0.117     | 0.127            | 0.109            | 0.111       |
> > > | Mean of Means     | 0.103     | 0.090     | 0.088          | 0.090     | 0.102     | 0.105            | 0.092            | 0.088       |
> > > | Ours (Profiles)   | 0.062     | **0.041** | 0.055          | 0.075     | 0.064     | 0.045            | 0.055            | 0.052       |
> > > | Ours (Models)     | **0.053** | 0.043     | **0.045**      | **0.073** | **0.050** | **0.040**        | **0.046**        | **0.041**   |
> > > | *Test Ratio: 80%* | |    | | | | | | |
> > > | Global Mean       | 0.125     | 0.140     | 0.115          | 0.093     | 0.115     | 0.131            | 0.132            | 0.139       |
> > > | Mean of Means     | 0.109     | 0.119     | 0.097          | 0.094     | 0.099     | 0.109            | 0.114            | 0.123       |
> > > | Ours (Profiles)   | 0.100     | **0.089** | 0.099          | **0.090** | 0.096     | 0.082            | 0.111            | 0.117       |
> > > | Ours (Models)     | **0.090** | 0.094     | **0.081**      | 0.092     | **0.088** | **0.075**        | **0.092**        | **0.095**   |
> > >
> > > We will update these experiments into our paper or the supplementary materials.

---

> > > ### Author Response · Authors · 2024-11-18
> > > **Reply 2 - Clarify the contributions and values of our work**
> > >
> > > > **The proposed method uses no actual predictions so it is risky.**
> > >
> > > We would like to highlight that our method is **not** based on "no actual predictions" and is not "a zero-shot manner". We predict unknown performance scores **based on known ones**.
> > >
> > > For example, when we want to predict the performance of model A on benchmark $\alpha$, we utilize some results of A on other benchmarks, and some of other models on $\alpha$. When testing model A on other tasks, we learn more about the "abilities" of A. When testing other models on $\alpha$, we gain some information on the properties of the benchmark $\alpha$, such as what abilities are evaluated on the benchmark. The information is the foundation for us to predict unknown performance.
> > >
> > > Second, in Section 3.5, we introduce model and dataset profiles, such as the vision encoder of the model, the number of parameters of the model, or the hidden representation cluster of a dataset. These profiles provide extra information about models and datasets for performance prediction.
> > >
> > > Last, our method provides uncertainty for performance prediction. As shown in Figure 4, uncertainties are correlated with the actual absolute errors and can inform us which predictions are not reliable.
> > >
> > > > **Contributions of the paper. Is the paper an advancement over the TinyBenchmark?**
> > >
> > > We respectfully highlight that our framework and TinyBenchmark are working on different problems and using different methodologies. Outperforming TinyBenchmark is not our goal. Our paper formulates a new problem, sheds light on correlation across models and benchmarks, and (in section 3.5) can capture the effects of different model or dataset profiles, which has not been done in previous works.

---

> > > > ### Comment · Reviewer_dktp · 2024-11-20
> > > >
> > > > I remain unconvinced about relying solely on zero-shot evaluations based on statistical patterns of other benchmarks without any testing, as this carries significant risks. Therefore, I believe we should conduct at least a few-shot evaluations while accounting for variables that various evaluation methods could introduce. Therefore, I will maintain my score.

---

> > > > > ### Author Response · Authors · 2024-11-23
> > > > > **Reply 2 - Further thoughts on our contributions**
> > > > >
> > > > > > **Further thoughts on our contributions**
> > > > >
> > > > > We are not claiming to replace direct evaluation. There is risk associated with using statistical results rather than the true benchmark, but there is also value of our approach in the development of new models and benchmarks.
> > > > >
> > > > > First, our approach estimates the correlation of benchmarks, which is particularly important given Reviewer dktp's comments on emerging LVLM benchmarks. As many new benchmarks are created, it is important to determine if they offer new insights into model performance or simply repeat what previous evaluation show. Our method estimates the correlation between model performances, helping to identify how much extra information a new benchmark may add.
> > > > >
> > > > > Second, we can reduce the evaluation cost in the development of new models and benchmarks. Let's say I want to develop LLaVA v3.0. We may need to try different designs and training methods. It is very expensive to evaluate model checkpoints on various tasks in each iteration of model development. We provide a tool for reducing the cost of evaluation and speeding up LVLM development. The final model performance can be determined by direct evaluation on each benchmark.
> > > > >
> > > > > In summary, our paper formulates a new problem, sheds light on correlation across models and benchmarks, and can reduce the unnecessary cost in model and benchmark development.

---

> > > > > ### Author Response · Authors · 2024-11-26
> > > > >
> > > > > Dear Reviewer dktp,
> > > > >
> > > > > Thank you for your valuable time and efforts in reviewing our paper!
> > > > >
> > > > > We would like to kindly remind you that we have conducted experiments combining few-shot evaluation with our method, which show significant improvement compared to using few-shot alone. We have also added arguments to clarify our contributions.
> > > > >
> > > > > If you have any further concerns or questions, please let us know, and we will be happy to address them. We look forward to your feedback.
> > > > >
> > > > > Best regards,
> > > > >
> > > > > The Authors

---

> > > > > > ### Comment · Reviewer_dktp · 2024-11-28
> > > > > >
> > > > > > I greatly appreciate the authors' efforts in conducting additional experiments. As you mentioned, predicting model performance solely based on the correlation of benchmark results may be meaningful in itself. However, compared to the existing problem of finding a minimal test set that is actionable and more robust, I believe this approach and methodology are too regressive. Perhaps the authors could not find this problem's unique benefits, so they did not present a direct rationale and instead added few-shot experiments.
> > > > > >
> > > > > > Furthermore, although a very simple heuristic was added to the few-shot experiments, I have serious doubts about whether this is truly novel in terms of methodology. It might be included as a simple analysis or discussion in the paper, though.
> > > > > >
> > > > > > I still think this paper's problem awareness and methodological contribution are very weak, so I will maintain my score.

---

> > > > > > > ### Author Response · Authors · 2024-12-04
> > > > > > >
> > > > > > > We sincerely thank Reviewer dktp for their thoughtful and engaging discussion. We firmly believe that our work offers significant value to the field of large vision-language models.

---

> ### Author Response · Authors · 2024-11-23
> **Reply 1 - Combine few-shot evaluation with our method**
>
> Thank you for your feedback! Below, we address your concern in detail.
>
> > **Combine few-shot evaluation with our method.**
>
> Previous studies ("few-shot evaluation" in Reviewer dktp's comment) select a small set of representative samples (coreset) in a benchmark and evaluate models on the coreset. Our work uses known model performance from different benchmarks or models for performance prediction. Our method is complementary to these existing approaches and can be combined with their few-shot evaluation.
>
> In experiments, we explore two combination methods:
>
> (1) Avg. Simply get the average prediction of few-shot evaluation and PMF;
>
> (2) Unc. Use uncertainties from MCMC to combine few-shot evaluation and PMF predictions. In short, when PMF is confident, we mainly rely on using known performance for prediction. Otherwise, the prediction is more dependent on few-shot evaluations
>
> Following LMMs-Eval, we use CLIP to generate embeddings for images and BGE-M3 for text, and concatenates them to create the final embeddings. Based on sample embeddings, we use random, Herding, and K-Center Greedy [5] to select core samples.
>
> The following table presents the average RMSE values of 3 experiments, demonstrating the effectiveness of our approach.
>
> | Method | Overall RMSE   | Overall MAE | Acc RMSE | Acc MAE | BART RMSE | BART MAE |
> | ------- | -------------- | ----------- | -------- | ------- | --------- | -------- |
> | *Select 5% samples*          | | | | | | |
> | Ours                         | 0.193          | 0.090       | 0.074    | 0.052   | 0.459     | 0.299    |
> | Random Selection             | 0.345          | 0.224       | 0.250    | 0.175   | 0.652     | 0.494    |
> | Random + Ours (Avg)          | 0.199 (-0.146) | 0.126       | 0.131    | 0.093   | 0.404     | 0.306    |
> | Random + Ours (Unc)          | 0.157 (-0.188) | 0.083       | 0.070    | 0.050   | 0.365     | 0.261    |
> | Herding                      | 0.326          | 0.220       | 0.252    | 0.177   | 0.582     | 0.458    |
> | Herding + Ours (Avg)         | 0.192 (-0.134) | 0.124       | 0.133    | 0.094   | 0.377     | 0.287    |
> | Herding + Ours (Unc)         | 0.155 (-0.171) | 0.081       | 0.070    | 0.050   | 0.358     | 0.252    |
> | K-Center Greedy              | 0.353          | 0.231       | 0.262    | 0.182   | 0.656     | 0.498    |
> | K-Center Greedy + Ours (Avg) | 0.200 (-0.153) | 0.128       | 0.137    | 0.096   | 0.394     | 0.302    |
> | K-Center Greedy + Ours (Unc) | 0.154 (-0.199) | 0.082       | 0.070    | 0.050   | 0.356     | 0.258    |
> | | | | | | | |
> | *Select 10% samples*         | | | | | | |
> | Ours                         | 0.193          | 0.090       | 0.074    | 0.052   | 0.459     | 0.299    |
> | Random Selection             | 0.224          | 0.141       | 0.152    | 0.107   | 0.444     | 0.326    |
> | Random + Ours (Avg)          | 0.149 (-0.075) | 0.088       | 0.085    | 0.061   | 0.322     | 0.237    |
> | Random + Ours (Unc)          | 0.139 (-0.085) | 0.076       | 0.069    | 0.049   | 0.313     | 0.224    |
> | Herding                      | 0.216          | 0.140       | 0.155    | 0.112   | 0.410     | 0.297    |
> | Herding + Ours (Avg)         | 0.144 (-0.072) | 0.088       | 0.087    | 0.064   | 0.305     | 0.220    |
> | Herding + Ours (Unc)         | 0.140 (-0.076) | 0.076       | 0.070    | 0.049   | 0.315     | 0.220    |
> | K-Center Greedy              | 0.223          | 0.142       | 0.154    | 0.109   | 0.437     | 0.322    |
> | K-Center Greedy + Ours (Avg) | 0.144 (-0.079) | 0.088       | 0.086    | 0.063   | 0.306     | 0.226    |
> | K-Center Greedy + Ours (Unc) | 0.138 (-0.085) | 0.077       | 0.070    | 0.049   | 0.313     | 0.224    |
> | | | | | | | |
> | *Select 15% samples*         | | | | | | |
> | Ours                         | 0.193          | 0.090       | 0.074    | 0.052   | 0.459     | 0.299    |
> | Random Selection             | 0.180          | 0.114       | 0.125    | 0.087   | 0.352     | 0.261    |
> | Random + Ours (Avg)          | 0.133 (-0.047) | 0.078       | 0.073    | 0.053   | 0.291     | 0.212    |
> | Random + Ours (Unc)          | 0.132 (-0.048) | 0.074       | 0.068    | 0.049   | 0.295     | 0.210    |
> | Herding                      | 0.177          | 0.117       | 0.130    | 0.093   | 0.332     | 0.245    |
> | Herding + Ours (Avg)         | 0.131 (-0.046) | 0.078       | 0.076    | 0.056   | 0.282     | 0.199    |
> | Herding + Ours (Unc)         | 0.135 (-0.042) | 0.074       | 0.069    | 0.049   | 0.302     | 0.210    |
> | K-Center Greedy              | 0.172          | 0.111       | 0.123    | 0.087   | 0.331     | 0.239    |
> | K-Center Greedy + Ours (Avg) | 0.129 (-0.043) | 0.076       | 0.073    | 0.053   | 0.281     | 0.203    |
> | K-Center Greedy + Ours (Unc) | 0.131 (-0.042) | 0.074       | 0.069    | 0.049   | 0.291     | 0.208    |
>
> [5] DeepCore: A Comprehensive Library for Coreset Selection in Deep Learning. https://arxiv.org/abs/2204.08499.

---

### Official Review · Reviewer_uZXr · 2024-10-31

**Soundness:** 3
**Presentation:** 3
**Contribution:** 3
**Rating:** 6
**Confidence:** 5

**Summary:**

Evaluating VLMs across various number of tasks is costly (as the number of benchmarks can be huge) and the model sizes can be very large as well. The paper tries to propose an approach to estimate the performance on some datasets, by converting the problem to that of sparse matrix factorization, a well studied statistical approach for matrix completion. They assume a M x N matrix, where M is the different models and N is the various tasks. Given some entries of this matrix, one can estimate the rest using matrix factorization. The paper proposes some trivial modifications to the standard PMF to fit this specific use case. While the proposed work is an application of existing techniques to this problem, it is unique and has not been done previously in this setting. The empirical results are great, and the proposed idea can be useful to the community as such, especially while practitioners are developing models and need to frequently evaluate a lot of checkpoints/variations/finetuned versions of VLMs.

**Strengths:**

- An interesting application of existing statistical method for the problem of estimating performance on benchmarks.
- The work has potential for impact and being useful for the community, especially developers.
- Easy to implement, nice and thorough empirical analysis with sufficient ablations and insights/discussions.

**Weaknesses:**

- In the active evaluation, the authors order the priority of the task for evaluation based on its estimation uncertainty/deviation. But this doesn’t factor in the cost of evaluation (time) or the model size for that entry. It can be possible that estimating 2 other entries with lower uncertainty initially, and a lower combined evaluation cost turns out to be better than evaluating the entry with highest uncertainty. Curious to know if the authors explored multi-objective optimization, or tried to incorporate evaluation cost in other versions of there proposed approach.
- As such, the work is basically applying an existing statistical technique (matrix completion) to the problem of estimating performance on benchmarks. Authors do propose some small modifications over standard matrix factorization. One can say that using matrix completion for various applications in the real world is not a novel contribution.
- It would have been much more compelling work, if the approach incorporated VLM specific ideas or benchmark specific stuff over an above the standard matrix factorization techniques.

**Questions:**

See the weakness section

---

> ### Author Response · Authors · 2024-11-14
> **Reply - Incorporating Evaluation Cost, Novelty, and VLM Specific Ideas**
>
> >  **Incorporate evaluation cost into the framework.**
>
> Thank you for the suggestion! It is very interesting to incorporate evaluation cost into our framework.
>
> A straightforward way is to implement a cost-aware heuristic function in active evaluation. Instead of using only uncertainty, we assign a score to each model-dataset pair, $score = f(uncertainty, cost)$. The model-dataset pairs with higher scores will be prioritized for evaluation. Some possible functions are $f(a, b)=a^\gamma b^{1-\gamma}, f(a, b)=a + \gamma b$.
>
> It is not easy to measure evaluation cost. Different models may have different acceleration techniques, some models are API-only, and evaluation samples may have various context lengths. Here, for each dataset, we simply use the number of samples / the total number of samples of all datasets to approximate the evaluation cost, and conduct a preliminary experiment.
>
> We follow the setting recommended by Reviewer Q3Aj. In short, we use 20% performance scores for initial training, 60% as the pool set, and 20% for testing. PMF is trained on the initial 20% data. In each iteration, we use a method to order the pool set and select the top model-dataset pairs. We retrain PMF in each iteration with extra data from pool set, and evaluate the model on the test set.
>
> The following table shows the RMSE Improvement (%) on the test set, where each column is the number of extra evaluated samples. If a method chooses more samples in each iteration, we fit the result into the closest column. As seen, our methods show significant improvement over Random.
>
> | Method             | 411k  | 823k  | 1234k | 1647k | 2058k | 2470k | 2881k (All) |
> | ------------------ | ----- | ----- | ----- | ----- | ----- | ----- | ----------- |
> | Random             | 7.2   | 11.2  | 14.7  | 17.9  | 20.98 | 23.06 | 23.00       |
> | Uncertainty - Cost | 20.85 | 23.28 | 23.04 | 22.26 | 22.53 | 23.25 | 23.00       |
> | Uncertainty / Cost | 19.44 | 23.44 | 23.84 | 22.39 | 22.34 | 22.74 | 23.00       |
>
> Interestingly, we find that, the error of performance prediction is mainly caused by small datasets, which may have large variance in performance due to limited sample size. Thus, evaluate models on these highly-uncertain but low-cost datasets may be the best for performance prediction. It is worth exploring more practical design and we note that our framework is extensible for that.
>
>
>
> > **Not a novel contribution**
>
> Thank you for acknowledging we are addressing an important problem. We respectfully highlight that the main contributions of our paper are: (1) formulating the problem of LVLM performance prediction based on known performance scores; and (2) connecting well-established algorithms to the novel application and demonstrating their effectiveness. Our main focus is the new application, instead of introducing technical contributions on a previous problem.
>
> > **Incorporated VLM specific ideas or benchmark specific stuff.**
>
> Thank you! We respectfully highlight that we incorporate model and dataset profiles into PMF (Section 3.5). For models, we include features such as the number of parameters in the LLM backbone, vision encoder type, and the LVLM family. Additionally, we explore three different approaches to generate these latent representations and cluster the LVLM benchmarks. These profiles may consider VLM- or benchmark-specific stuff as you expected, are extensible for extra model or dataset information, and improve the accuracy in performance prediction.

---

> > ### Comment · Reviewer_uZXr · 2024-11-19
> > **Thanks for the response**
> >
> > I thank the reviewers for addressing my concerns. I will maintain my score.

---

> > > ### Author Response · Authors · 2024-11-20
> > >
> > > Thank you for your feedback! We really appreciate your support and for keeping the score. If you have any other questions or suggestions, we’re happy to discuss them with you.

---

### Official Review · Reviewer_VvNb · 2024-11-04

**Soundness:** 3
**Presentation:** 2
**Contribution:** 2
**Rating:** 6
**Confidence:** 3

**Summary:**

This paper introduces a framework for predicting the performance of large vision-language models (LVLMs) across multiple tasks. The main idea is to employ probabilistic matrix factorization (PMF) to estimate unknown performance scores based on a sparse set of observed scores. By formulating performance prediction as a matrix completion problem and leveraging MCMC methods to estimate prediction uncertainty, the authors aim to reduce the computational cost of evaluating large models across diverse tasks. In addition, the authors propose several enhancements to handle data sparsity, including tensor factorization for multiple performance metrics and Bayesian PMF.

**Strengths:**

1. The proposed method is grounded in well-established techniques of matrix factorization and probabilistic modeling. The mathematical foundation of PMF is solid, and using MCMC for uncertainty estimation is a sensible approach to prioritize evaluations. The paper demonstrates that the method can effectively predict unknown performance scores, especially when more than 10% of the data is available.

2. Evaluating 108 LVLMs across 176 datasets demonstrates the practicality and scalability of the proposed method across a wide range of tasks.

**Weaknesses:**

1. The paper tackles an important problem: efficiently evaluating large-scale models as they grow in size and complexity. The idea of using matrix completion and active evaluation is interesting and, if successful, could lead to significant computational savings. However, the novelty is somewhat limited since the approach mainly builds on existing techniques like PMF, Bayesian modeling, and MCMC.

2.  Several parts of the paper lack clear explanations. For example, the differences between PMF, PTF, and Bayesian PMF are densely presented, and their respective impacts on performance are not sufficiently disentangled in the experiments. An explicit ablation study would help understand each enhancement's individual contributions.

3. While using uncertainty to prioritize evaluations is compelling, the results show a gap between the uncertainty-based approach and the oracle method. The paper could explore why this gap exists and whether alternative heuristics could narrow it.

**Questions:**

1. Could you clarify how Bayesian PMF differs from standard PMF in practical terms? Specifically, how does incorporating an LKJ prior (Lewandowski et al., 2009) impact the predictions in practice?

2. Including an ablation study to better quantify the contribution of each component—such as tensor factorization, Bayesian PMF, and the use of profiles—would help clarify their respective impacts on performance.

3. There's a noticeable gap between the uncertainty-based active evaluation and the oracle method. Have you considered alternative heuristics for prioritizing evaluations that might close this gap?

---

> ### Author Response · Authors · 2024-11-14
> **Reply 1 - Novelty, Bayesian PMF, and LKJ prior**
>
> We sincerely appreciate your time and effort in reviewing our paper. We are pleased that you acknowledged the solid foundation, effectiveness, practicality, and scalability of our paper.
>
> > **The paper tackles an important problem but the novelty is somewhat limited.**
>
> Thank you for acknowledging we are addressing an important problem. We respectfully highlight the main contributions of our paper are (1) we formulate the problem of LVLM performance prediction based on known performance scores; and (2) we connect well-established algorithms to the novel application and show their effectiveness. Our main focus is the new application, instead of introducing technical contributions on a previous problem.
>
> > **How Bayesian PMF differs from standard PMF in practical terms?**
>
> In real-world situations, we may only have limited performance data about an LVLM or benchmark for training PMF. For example, if OpenAI released GPT-5 yesterday, we might know its performance on only 5 benchmarks. In cases like this, Bayesian PMF can predict performance more accurately than the standard PMF model, which is shown in Figure 5(A).
>
> The reason is that Bayesian PMF defines distributions over the parameters of prior distributions, known as hyperpriors. The hyperpriors work similarly to regularization terms in a loss function and improve model performance when there is limited data available. As we gather more data, the advantage of using hyperpriors over a standard model becomes less noticeable.
>
> > **How does incorporating an LKJ prior impact the predictions in practice?**
>
> Using an LKJ prior is primarily for computational reasons, rather than improving predictions. We will clarify this in the revised paper.
>
> In short, Wishart distribution models the distribution of covariance matrices. It has two main issues during the sampling process.
>
> * First, it requires the sampled matrices to be both positive-definite and symmetric. The probability of generating a valid sample by randomly changing elements is close to zero.
> * Second, the distribution has a very heavy tail, which poses many challenges for simple sampling methods.
>
> Instead, we use the LKJ correlation prior and an Exponential prior, which are computationally advantageous.
>
> ----
> Our references:
>
> This is suggested by PyMC Official Documentation.
>
> LKJ Cholesky Covariance Priors for Multivariate Normal Models. https://www.pymc.io/projects/examples/en/latest/howto/LKJ.html
>
> There are also discussions on practical issues about the Wishart distribution vs LKJ on coding forums like GitHub and StackExchange.
>
> https://github.com/pymc-devs/pymc3/issues/538#issuecomment-94153586

---

> ### Author Response · Authors · 2024-11-14
> **Reply 2 - Ablation Study and Uncertainty-Based Active Evaluation**
>
> > **Ablation study to better quantify the contribution of each component.**
>
> We conduct a detailed ablation study as suggested. The table below shows each component’s contribution when the training performance scores are sparse. Here, PMF methods model each metric separately. All results are the average RMSE over 10 experiments, with lower values indicating better performance.
>
> | Method                    | Overall            | Acc       | Precision | Recall    | F1        | BART      | BERT      |
> | ------------------------- | ------------------ | --------- | --------- | --------- | --------- | --------- | --------- |
> | *Test Ratio: 80%*         |                    |           |           |           |           |           |           |
> | PMF                       | 0.267 (+0.000)     | 0.115     | 0.205     | 0.237     | 0.197     | 0.707     | 0.085     |
> | PMF + Bayesian            | 0.254 (-0.013)     | 0.118     | 0.197     | 0.224     | 0.184     | 0.664     | **0.083** |
> | PMF + Profiles            | 0.254 (-0.013)     | 0.111     | 0.193     | 0.230     | 0.186     | 0.672     | 0.084     |
> | PTF                       | 0.249 (-0.018)     | 0.120     | 0.151     | **0.207** | **0.145** | 0.661     | 0.091     |
> | PTF  + Bayesian           | 0.239 (-0.028)     | 0.116     | 0.152     | 0.212     | 0.151     | 0.630     | 0.090     |
> | PTF  + Profiles           | 0.240 (-0.027)     | 0.115     | 0.151     | 0.208     | 0.147     | 0.637     | 0.088     |
> | PTF + Bayesian + Profiles | **0.236 (-0.031)** | **0.108** | **0.147** | 0.208     | 0.147     | **0.627** | 0.089     |
> | *Test Ratio: 90%*         |                    |           |           |           |           |           |           |
> | PMF                       | 0.327 (0.000)      | 0.160     | 0.238     | 0.261     | 0.227     | 0.862     | 0.096     |
> | PMF + Bayesian            | 0.296 (-0.031)     | 0.144     | 0.225     | 0.254     | 0.213     | 0.774     | **0.091** |
> | PMF + Profiles            | 0.313 (-0.014)     | 0.146     | 0.232     | 0.260     | 0.220     | 0.828     | 0.094     |
> | PTF                       | 0.294 (-0.033)     | 0.161     | 0.194     | 0.235     | 0.187     | 0.761     | 0.094     |
> | PTF + Bayesian            | **0.267 (-0.060)** | 0.142     | **0.179** | 0.232     | 0.178     | **0.690** | 0.093     |
> | PTF  + Profiles           | 0.274 (-0.053)     | 0.145     | 0.190     | 0.233     | 0.184     | 0.710     | 0.092     |
> | PTF + Bayesian + Profiles | **0.267 (-0.060)** | **0.138** | **0.179** | **0.228** | **0.176** | 0.698     | 0.093     |
>
> Some results are already presented in Table 1 and Figure 5 in the paper. We will include the detailed results in the supplementary materials.
>
> > **Gap between the uncertainty-based active evaluation and the oracle method.**
>
> We implement a canonical active learning evaluation setup, as suggested by Reviewer Q3Aj. In short, we use 20% performance scores for initial training, 60% as the pool set, and 20% for testing. PMF is trained on the initial 20% data. In each iteration, we use Random / Active (Ours) / Oracle methods to order the pool set and select the top model-dataset pairs. We retrain PMF in each iteration with extra data from pool set, and evaluate the model on the test set.
>
> The following table reports the improvement on the test set as we acquire more evaluations in the pool set.
>
> | Method            | 0%    | 5%    | 10%   | 20%   | 30%   | 40%   | 50%   | 60% (All) |
> | ----------------- | ----- | ----- | ----- | ----- | ----- | ----- | ----- | --------- |
> | *RMSE*            |       |       |       |       |       |       |       |           |
> | Random            | 0.258 | 0.247 | 0.236 | 0.224 | 0.215 | 0.205 | 0.197 | 0.195     |
> | Active (Ours)     | 0.258 | 0.220 | 0.207 | 0.196 | 0.192 | 0.192 | 0.192 | 0.195     |
> | Oracle            | 0.258 | 0.213 | 0.202 | 0.199 | 0.196 | 0.195 | 0.194 | 0.195     |
> | *Improvement (%)* |       |       |       |       |       |       |       |           |
> | Random            | 0.0   | 4.1   | 8.1   | 13.0  | 16.5  | 20.1  | 23.3  | 24.3      |
> | Active (Ours)     | 0.0   | 14.3  | 19.6  | 24.0  | 25.2  | 25.4  | 25.4  | 24.2      |
> | Oracle            | 0.0   | 17.4  | 21.6  | 22.7  | 23.7  | 24.2  | 24.4  | 24.3      |
>
> As shown in the table, our method demonstrates significant improvement over Random and approaches or even exceeds the performance of Oracle.
>
> Additionally, as suggested by Reviewer uZXr, we also explore cost-aware active evaluation and demonstrate the advantages of our methods. We kindly refer you to our reply to Reviewer uZXr.

---

> > ### Comment · Reviewer_VvNb · 2024-11-20
> >
> > Thanks for your detailed response, I have raised my score from 5 to 6.

---

> > > ### Author Response · Authors · 2024-11-20
> > >
> > > Thank you for your feedback! We really appreciate your support and for considering raising the score. If you have any other questions or ideas, we’re happy to discuss them with you.

---

### Official Review · Reviewer_Q3Aj · 2024-11-04

**Soundness:** 3
**Presentation:** 4
**Contribution:** 3
**Rating:** 8
**Confidence:** 4

**Summary:**

This paper provides a framework for predicting the performance of large vision-language models on held-out downstream tasks using a small set of observed task performances, i.e., evaluations for a small set of (model, dataset) tuples. They formulate this as a matrix completion problem and demonstrate that probabilistic matrix factorization (PMF) with MCMC is surprisingly effective, using a large set of 108 VLM evaluations on 176 datasets. Further, the authors demonstrate that the uncertainty estimates of PMF can be used in active evaluation to prioritize which evaluation to conduct next, outperforming random selection. Lastly, the work explores extensions of the naive Bayesian PMF model: tensor factorization to handle multiple metrics, and incorporating side information for better performance under extreme sparsity.

**Strengths:**

- Strong motivation: VLM evaluation is very expensive, so being able to accurately predict downstream evaluation performance from a limited set of evaluations is very valuable.
- The method is elegant and appears to work well, e.g., the correlation plots in Figure 3 look clean. It is also surprisingly effective in active evaluation, which is a very practical and exciting direction for this line of work.
- The paper is exceptionally well-written and clear.
- The evaluation uses a large set of (model, dataset) evaluations on a variety of open- and closed-source models.

**Weaknesses:**

The authors only consider a limited set of naive baselines for the main experiments in Figure 3. Could the authors benchmark other more sophisticated (neural) matrix completion methods, such as deep matrix factorization [1] or Graph Convolutional Matrix Completion [2]?

[1] Arora et al., 2019. Implicit Regularization in Deep Matrix Factorization. In NeurIPS. https://arxiv.org/abs/1905.13655

[2] van den Berg et al., 2018. Graph Convolutional Matrix Completion. In KDD. https://www.kdd.org/kdd2018/files/deep-learning-day/DLDay18_paper_32.pdf

**Questions:**

* My main concern is the limited set of baseline matrix completion methods (mentioned above).
* Evaluation of active evaluation: could you consider a more canonical active learning evaluation setup? i.e., randomly partition elements of the matrix into an initial training set, an "unlabeled pool set" (in the active learning nomenclature), and a test set, and report active learning-style curves: for each acquisition method (oracle, random, uncertainty), plot RMSE on the test set versus the number of acquisition steps, as you acquire evals in the pool set? e.g., what is done in [3].
* Comment for possible future work: because the indices of the unobserved (model, dataset) elements are known a priori (and you also have access to side information such as which image encoder was used, etc.), this setting seems to fit naturally with some transductive active learning methods, such as [4].

[3] Gal et al., 2017. Deep Bayesian Active Learning with Image Data. https://arxiv.org/abs/1703.02910

[4] Bickford-Smith et al., 2023. Prediction-Oriented Bayesian Active Learning. In AISTATS. https://proceedings.mlr.press/v206/bickfordsmith23a/bickfordsmith23a.pdf

---

> ### Author Response · Authors · 2024-11-14
> **Reply 1 - More Baseline Methods**
>
> We greatly appreciate your time and effort in reviewing our paper. We are pleased that you acknowledged the strong motivation, elegance, effectiveness, clear writing, and comprehensive evaluation of our work.
>
> > **The limited set of baseline matrix completion methods.**
>
> As you suggested, we evaluate Deep Matrix Factorization (DMF) [1] and Graph Convolutional Matrix Completion (GCMC) [2] for a more comprehensive comparison.
>
> For DMF, we use MSE loss and the Adam optimizer. The learning rate is 1e-3 and the batch size is 256. The embedding dimension of each user or item is 10, which is the same for PMF. We train DMF for 200 epochs and the result of the best epoch is reported.
>
> For GCMC, we refer to the GitHub implementation (https://github.com/riannevdberg/gc-mc). Dropout ratio is 0.7, learning rate is 0.01, hidden units are [500, 75] in 1st and 2nd layers, accumulation method is "stack", the number of basis functions is 2, and the model is trained for 200 epochs. We note that there are two main issues when using GCMC:
>
> - GCMC handles discrete rating levels and treats each rating level as a separate class (see Section 2.3 [2]), which is not suitable for our setting, because we use continuous ratings like the BART  scores. To address this, we only use LVLM benchmarks with accuracy as the main metric, and discretize accuracy into 101 classes, i.e., {0, ..., 100}.
> - When training data is sparse, some classes (for example, accuracy = 67) do not occur in training set, leading to running errors in the code.
>
> The table below summarizes the average results from 10 experiments. As shown, PMF demonstrates superior performance on our dataset compared to DMF and GCMC.
>
> | Method            | Overall RMSE | Overall MAE | Acc RMSE | Acc MAE | BART RMSE | BART MAE |
> | ----------------- | ------------ | ----------- | -------- | ------- | --------- | -------- |
> | *Test ratio: 20%* |              |             |          |         |           |          |
> | DMF [1]           | 0.225        | 0.105       | 0.086    | 0.060   | 0.538     | 0.353    |
> | GCMC [2]          |              |             | 0.187    | 0.139   |           |          |
> | PMF               | 0.193        | 0.090       | 0.074    | 0.052   | 0.461     | 0.303    |
> | *Test ratio: 80%* |              |             |          |         |           |          |
> | DMF [1]           | 0.561        | 0.314       | 0.289    | 0.209   | 1.26      | 0.896    |
> | GCMC [2]          |              |             | -        | -       |           |          |
> | PMF               | 0.317        | 0.174       | 0.156    | 0.115   | 0.723     | 0.504    |
>
> We notice that matrix completion methods are commonly applied in recommender systems, where there are usually thousands of users and items, with millions of samples, such as the Movielens dataset. But in our setting, we have 108 models, 176 datasets, and 19K samples. Thus, we build our method on a simple but strong baseline, PMF, rather than neural networks, which are possibly more data-hungry.

---

> ### Author Response · Authors · 2024-11-14
> **Reply 2 - Active Learning Evaluation and Transductive Active Learning**
>
> > **A more canonical active learning evaluation setup.**
>
> Thank you for the suggestion! We use 20% data for initial training, 60% as the pool set, and 20% for testing. The following table reports the improvement on the test set as we acquire more evaluations in the pool set.
>
> | Method            | 0%    | 5%    | 10%   | 20%   | 30%   | 40%   | 50%   | 60% (All) |
> | ----------------- | ----- | ----- | ----- | ----- | ----- | ----- | ----- | --------- |
> | *RMSE*            |       |       |       |       |       |       |       |           |
> | Random            | 0.258 | 0.247 | 0.236 | 0.224 | 0.215 | 0.205 | 0.197 | 0.195     |
> | Active (Ours)     | 0.258 | 0.220 | 0.207 | 0.196 | 0.192 | 0.192 | 0.192 | 0.195     |
> | Oracle            | 0.258 | 0.213 | 0.202 | 0.199 | 0.196 | 0.195 | 0.194 | 0.195     |
> | *Improvement (%)* |       |       |       |       |       |       |       |           |
> | Random            | 0.0   | 4.1   | 8.1   | 13.0  | 16.5  | 20.1  | 23.3  | 24.3      |
> | Active (Ours)     | 0.0   | 14.3  | 19.6  | 24.0  | 25.2  | 25.4  | 25.4  | 24.2      |
> | Oracle            | 0.0   | 17.4  | 21.6  | 22.7  | 23.7  | 24.2  | 24.4  | 24.3      |
>
> As shown in the table, our method shows significant improvement over Random, and is close to Oracle. Interestingly, when we acquire around half of the pool set, the model shows better performance than using the entire pool set. We will use this setup and update related results in our paper.
>
> > **Extension to transductive active learning.**
>
> Thank you! It is very interesting to explore transductive active learning within our framework. In practice, we might ask questions like, "What evaluation experiments can best inform the performance of models that use CLIP as vision encoders?" or "Which experiments provide the most useful information for improving our own models?" In such cases, instead of looking at uncertainties across all predictions, it could be more helpful to measure the information gain to a specific model or dataset.
>
> An intuitive approach would be to integrate the expected predictive information gain (EPIG) method proposed by Bickford-Smith et al. [4] into our framework. This idea diverges from our current focus and would be better suited for future work.

---

> ### Comment · Reviewer_Q3Aj · 2024-11-15
> **Response to Author Rebuttal**
>
> Thanks for your thorough response addressing my concerns on lack of deep baselines, and adding the active learning experiment in the more canonical setting. I have raised my score from 6 to 8.

---

> > ### Author Response · Authors · 2024-11-15
> >
> > Thank you for your feedback! We truly appreciate your support and your consideration in raising the score. If you have any more questions or suggestions, we are happy to discuss them with you.

---

### Author Response · Authors · 2024-12-04

We sincerely appreciate all the reviewers for their thoughtful engagement in the discussion and their valuable input into our work.

---

### Meta-Review · Area_Chair_7Dgt · 2024-12-19

**Metareview:**

This paper proposes a new framework for predicting unknown performance scores based on observed ones from other LVLMs or tasks. This paper formulates the performance prediction as a matrix completion task and applies probabilistic matrix factorization with Markov chain Monte Carlo to solve this problem. This paper also introduces several improvements to enhance probabilistic matrix factorization for scenarios with sparse observed performance scores. Experiments are conducted to demonstrate the effectiveness of the proposed method.

Pros:

- The studied problem is interesting and practically important.
- This paper covers a comprehensive list of vision-language models/tasks.

Reasons to reject:

- Simply formulating the problem of LVLM performance prediction as a matrix completion problem seems not reasonable. It is a purely statistical problem if we do so, without any additional information (i.e., a small part of the test dataset).

- To solve this matrix completion problem, this paper did not propose any new method or involve the features or properties of large vision-language models (LVLMs) or vision-language tasks. I feel that even the model is a linear model or other deep models, with general tasks (instead of vision-language tasks), the matrix completion problem can be still formulated. So I think it is important to indicate why the proposed method of this paper is suitable for vision-language scenarios. From my side, I did not find any vision-language information leveraged in this paper.

**Additional Comments On Reviewer Discussion:**

This paper finally receives the scores of 8 (Reviewer Q3Aj), 6 (Reviewer VvNb), 6 (Reviewer uZXr), and 3 (Reviewer dktp). Reviewer dktp still votes for rejection and indicates that the motivation of this paper is fundamentally flawed and this paper has no technical novelty. I agree with Reviewer dktp, as stated in my above reasons to reject.

---

### Decision · Program_Chairs · 2025-01-22

Reject